# IFT trains in different stages of assembly queue at the ciliary base for consecutive release into the cilium

Jenna L Wingfield[1], Ilaria Mengoni[1], Heather Bomberger[1,2†], Yu-Yang Jiang[1], Jonathon D Walsh[3], Jason M Brown[4,5], Tyler Picariello[4], Deborah A Cochran[4], Bing Zhu[6], Junmin Pan[6,7], Jonathan Eggenschwiler[3], Jacek Gaertig[1], George B Witman[4], Peter Kner[2], Karl Lechtreck[1]*

[1]Department of Cellular Biology, University of Georgia, Athens, United States; [2]College of Engineering, University of Georgia, Athens, United States; [3]Department of Genetics, University of Georgia, Athens, United States; [4]Department of Cell and Developmental Biology, University of Massachusetts Medical School, Worcester, United States; [5]Department of Biology, Salem State University, Salem, United States; [6]MOE Key Laboratory of Protein Sciences, Tsinghua-Peking Center for Life Sciences, School of Life Sciences, Tsinghua University, Beijing, China; [7]Laboratory for Marine Biology and Biotechnology, Qingdao National Laboratory for Marine Science and Technology, Qingdao, China

*For correspondence: lechtrek@uga.edu

Present address: [†]Department of Biomedical Engineering, University of Minnesota, Minneapolis, United States

Competing interests: The authors declare that no competing interests exist.

**Abstract** Intraflagellar transport (IFT) trains, multimegadalton assemblies of IFT proteins and motors, traffic proteins in cilia. To study how trains assemble, we employed fluorescence protein-tagged IFT proteins in *Chlamydomonas reinhardtii*. IFT-A and motor proteins are recruited from the cell body to the basal body pool, assembled into trains, move through the cilium, and disperse back into the cell body. In contrast to this 'open' system, IFT-B proteins from retrograde trains reenter the pool and a portion is reused directly in anterograde trains indicating a 'semi-open' system. Similar IFT systems were also observed in *Tetrahymena thermophila* and IMCD3 cells. FRAP analysis indicated that IFT proteins and motors of a given train are sequentially recruited to the basal bodies. IFT dynein and tubulin cargoes are loaded briefly before the trains depart. We conclude that the pool contains IFT trains in multiple stages of assembly queuing for successive release into the cilium upon completion.

## Introduction

The assembly of most cilia and flagella (terms used interchangeably) depends on bidirectional intra-flagellar transport (IFT) (*Rosenbaum and Witman, 2002*). Anterograde IFT trains move from the cili-ary base to the tip powered by kinesin-2; in retrograde IFT, the trains return to the cell body employing IFT dynein. The IFT trains transport proteins in and out of cilia to support ciliary assembly, maintenance, and signaling (for review: [*Lechtreck, 2015*]). They are composed of IFT motors and IFT particles, the latter consisting of 22 conserved IFT proteins organized into biochemically stable IFT-A, IFT-B1 and IFT-B2 subcomplexes each consisting of equimolar amounts of 6, 10 and 6 pro-teins, respectively (*Cole et al., 1998*; *Katoh et al., 2016*; *Taschner et al., 2012*, *2016*). At the ultra-structural level, the trains possess a periodic substructure presumably resulting from the 22-subunit IFT particles (*Pigino et al., 2009*; *Stepanek and Pigino, 2016*; *Vannuccini et al., 2016*). Numerous protein-protein interactions within the IFT particle have been identified and a 15-subunit IFT-B

complex has been assembled from recombinant proteins (*Hirano et al., 2017*; *Taschner et al., 2016*). The oligomerization of purified IFT complexes to the size of trains observed in vivo under cell free conditions has not yet been reported. In summary, our knowledge on how trains assemble is limited.

Western blot analyses showed that 90% or more of the IFT proteins reside in the cell body, a portion of which is concentrated around the basal bodies forming an IFT pool (*Ahmed et al., 2008*; *Deane et al., 2001*; *Richey and Qin, 2012*). The IFT trains emerge from the basal body pool (bb-pool) and enter the cilia in a kinesin-2 dependent manner (*Kozminski et al., 1995*). In contrast to the discrete IFT trains observed inside cilia, the IFT bb-pool appears to be amorphous at the ultra-structural level, i.e., bona fide IFT trains with their characteristic zigzag structure are visible only near the distal end of the basal bodies (*Rogowski et al., 2013*). IFT-B loss-of-function mutants often fail to assemble cilia presumably due to impaired IFT train formation. In such mutants, the stability of the remaining IFT proteins and their spatial distribution in the bb-pool are altered (see for example: [*Hou et al., 2007*; *Lv et al., 2017*; *Richey and Qin, 2012*]). Studies on such mutants clarify the interdependence among IFT proteins during subcomplex assembly and recruitment into the bb-pool. They are, however, less suited to determine the dynamics of IFT proteins in the bb-pool during the assembly of functional IFT trains. In a pioneering study, Buisson and colleagues used GFP-tagged IFT52 and fluorescence recovery after photobleaching (FRAP) to study the assembly of IFT trains in *Trypanosoma brucei* (*Buisson et al., 2013*). After photobleaching of the bb-pool, anterograde traffic of fluorescent IFT trains was transiently interrupted, indicative of bleached trains exiting the bb-pool. The pool signal recovered partially and anterograde IFT traffic resumed albeit with reduced signal strength suggesting that photobleached proteins in the bb-pool mix with unbleached IFT proteins supposedly derived from retrograde trains. This led to the proposal of a closed IFT system in which the IFT proteins perpetually cycle between the flagellum and the bb-pool with limited exchange of proteins between the flagellum-basal body entity and the rest of the cell body.

Here, we used a collection of eight fluorescent protein (FP)-tagged IFT proteins to study the assembly of IFT trains in *C. reinhardtii*, which allows for high quality imaging of protein traffic inside flagella by total internal reflection fluorescence (TIRF) microscopy (*Engel et al., 2009*; *Lechtreck, 2013a*, *Lechtreck, 2016*). Our analysis revealed a largely open IFT system in which proteins are continuously recruited from the cell body to the basal bodies, assembled into trains, and released back into the cell body after traveling through the cilium. Bleaching of the bb-pool was followed by a temporal gap in fluorescence anterograde IFT traffic of several seconds before unbleached trains reentered the cilium. Notably, the duration of this gap differed among the IFT proteins analyzed suggesting that IFT-A, IFT-B, and finally the motor proteins are sequentially recruited into assembling trains. Our data indicate a high temporospatial organization of the IFT bb-pool with IFT trains in various stages of assembly queuing near the basal bodies for sequential release into the cilium.

## Results

### IFT proteins occupy distinct regions of the basal body pool

To analyze the dynamics of IFT proteins in the pool surrounding each of the two flagella-bearing basal bodies of *Chlamydomonas reinhardtii*, we employed strains expressing fluorescent protein (FP)-tagged IFT particle and motor proteins (*Figure 1A*). IFT27 was expressed in a wild-type strain and KAP-GFP was expressed in *fla3*, which has a mutant KAP of reduced function (*Mueller et al., 2005*; *Qin et al., 2007*). The other proteins were expressed in loss-of-function mutants and rescued the corresponding flagella assembly phenotypes (*Figure 1A,B*). The transgenic strains had normal or nearly normal length flagella and displayed IFT traffic with mostly wild-type IFT velocities and anterograde frequencies (~1/s); retrograde frequencies were more variable which we attribute mostly to the weaker signals of retrograde trains (*Figure 1A–D*; *Video 1*). Trajectories representing retrograde traffic of KAP-GFP, which in *C. reinhardtii* mostly dissociates from IFT trains at the tip, were faint and infrequent. For our experiments, we selected cells displaying frequent IFT traffic avoiding cells with stationary IFT proteins accumulated inside the flagella (*Stepanek and Pigino, 2016*). To image IFT proteins in the basal body pool, the incident angle of the laser beam was increased to allow for a deeper penetration of the light into the specimen. The various FP-tagged IFT proteins were present

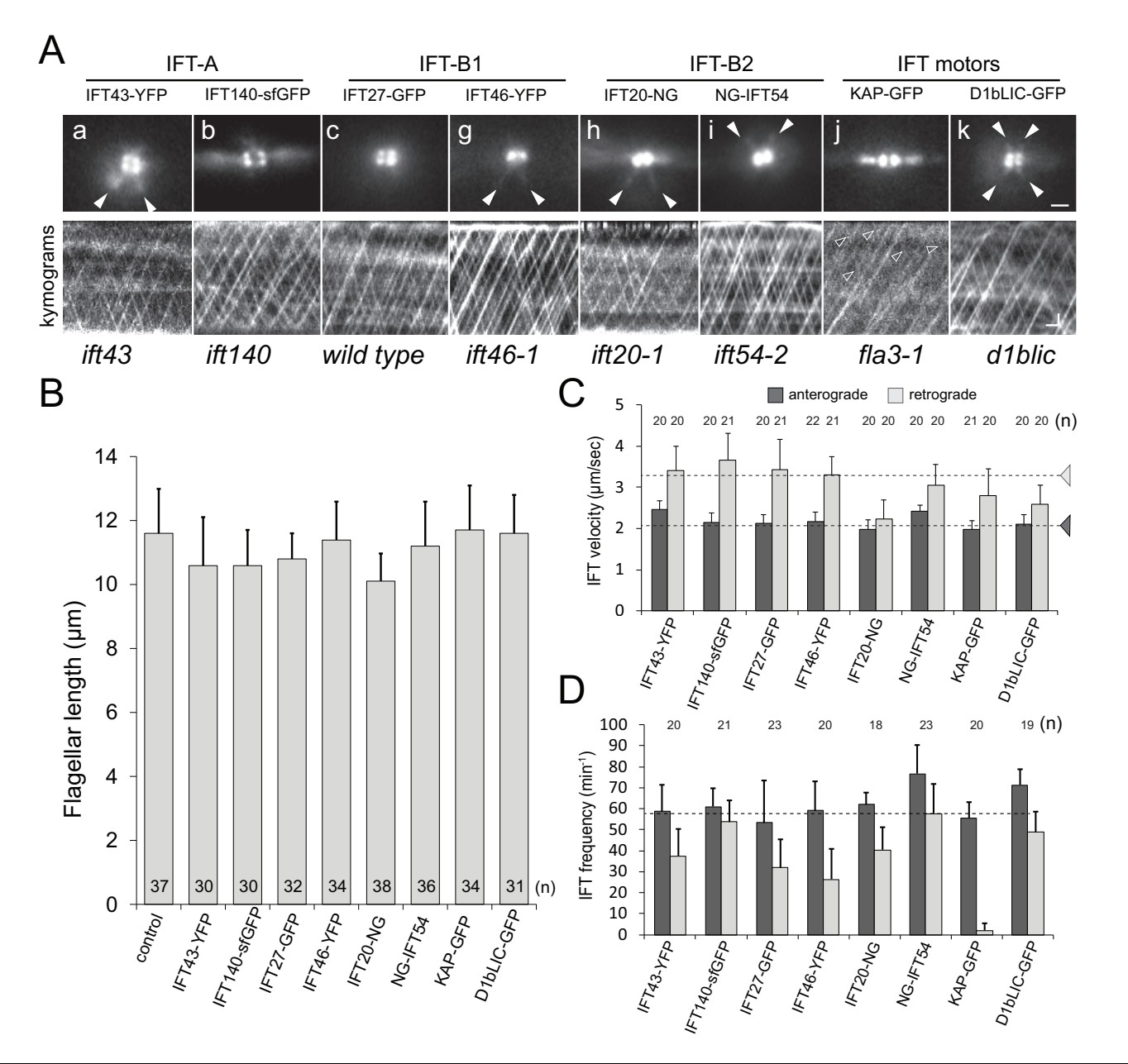

**Figure 1.** IFT proteins are concentrated at the flagellar base. (A) top: Still images from live cells showing the distribution of FP-tagged IFT proteins in the basal body pool. Arrowheads: IFT proteins along the microtubular roots. To enhance clarity, 10-frame image averages are shown. Bar = 1 μm. bottom: Kymograms showing the trajectories of IFT trains inside flagella. Anterograde traffic results in trajectories running from the bottom left to the top right; retrograde traffic is presented by lines running from the top left to the bottom right. The genetic background of the strains is indicated. Bars = 1 μm 1 s. (B) Flagellar length of the strains expressing FP-tagged IFT proteins. The standard deviation and the number of measurements are indicated. Strain CC-620 was used as a control. (C) Velocity of anterograde and retrograde IFT traffic in the rescue strains. Note the reduction in retrograde velocity of IFT20-NG. Dashed lines indicate the velocities of anterograde and retrograde trains as determined by DIC microscopy based of *Reck et al. (2016)*. (D) Frequency of IFT traffic. While the anterograde frequency was close to one train/s for all strains, retrograde frequencies were lower and more variable than reported for IFT in *C. reinhardtii*. The usage of fast bleaching fluorescent proteins (i.e., YFP) and the low laser intensity used here to allow for long-term imaging could have prevented the detection of weaker trains. The standard deviation is indicated.

**Video 1.** TIRF microscopy of NG-IFT54. *Chlamydomonas* cells tend to adhere with their two flagella to smooth surfaces including cover glass allowing for TIRF imaging of fluorescently labeled proteins inside the flagella. The approximate positions of the flagellar tips (tip1 and tip2) and of the cell body (CB) are indicated. Images were acquired at 10 fps and the video is displayed in real time. The timer counts mm:ss.

in dot-, colon-, or dumbbell-shaped regions near each basal body; most of them also showed weak X-shaped signals representing the microtubular flagellar roots (*Figure 1A*). Focal series from the flagella down into the cell body revealed that most of KAP-GFP was close to the distal end of the basal bodies followed by accumulations of the tagged IFT-B proteins; the IFT-A proteins and D1bLIC were concentrated deeper in the cell (not shown; [*Brown et al., 2015*]). The IFT140-sfGFP signals were more peripheral than those of most IFT-B proteins. Within the limitations of the technique used, live cell imaging confirms and extends pervious observations on fixed cells indicating that different IFT proteins inhabit non-identical spatial domains within the bb-pool (*Hou et al., 2007*).

## IFT proteins in the basal body pool exchange at different rates

A focused laser beam in epi-illumination was used to photobleach one of the two IFT pools at the flagellar base (*Figure 2A*, *Figure 2—figure supplement 1A–C*). The IFT bb-pool signals of all eight FP-tagged IFT particle and motor proteins recovered on average within 3 to 10 s (*Figure 2B*; *Video 2*). To quantify the degree and rate of recovery, we normalized the signal intensity of the bleached experimental bb-pool for fluorescence loss encountered by the unbleached control bb-pool due to the continuous TIRF illumination (*Figure 2C,D*). Signal recovery typically exceeded 60% of the prebleach strength and often complete recovery to the level of the control bb-pool was observed (*Figure 2C*). Recovery of the signal was also observed after repeated bleaching of the bb-pool indicating that the size of one IFT bb-pool is small as compared to the total cellular supply of IFT proteins (*Figure 2D*, *Figure 2—figure supplement 1D,E*). While the KAP-GFP signal recovered in less than 4 s, IFT43-YFP, IFT20-FP and NG-IFT54 required ~9s to reach maximum recovery (*Figure 2E*). In conclusion, IFT proteins in the bb-pool exchange at different rates.

## The dispatched IFT-A and motor proteins do not return to the basal body pool

The unbleached proteins causing the recovery of a bleached bb-pool could either be derived from retrograde IFT or newly recruited from the cell body (referred to here as the 'cell body pool' or cb-pool). To determine which source resupplies the IFT bb-pool, we used a fluorescence loss in photobleaching (FLIP) approach by placing a focused laser beam near the tip of one flagellum preventing the return of unbleached IFT proteins to the basal body via retrograde IFT (*Figure 3A*). The duty ratio of the bleaching laser was set to 100–200 ms on and 400–900 ms off and in many experiments, images were only recorded while the bleaching laser was off to eliminate over-exposed frames (compare *Figure 3B,C,F* and *Figure 3—figure supplement 1A*). TIRF imaging at the flagella level identified fluorescent retrograde IFT trains in the control flagellum but not in the experimental flagellum (*Figure 3B*, *Figure 3—figure supplement 1A*). Focusing onto the basal bodies, the fluorescence intensity of the bb-pool attached to the experimental flagellum was determined and normalized for general fluorescence loss using the control basal body. For the IFT-B proteins IFT20-NG, IFT46-YFP, and NG-IFT54 the signal intensity of the experimental basal body decreased significantly during FLIP illumination (*Figure 3C,D,F,G*). The fluorescence loss was substantially lower for the tagged IFT27, motor protein subunits, and in particular IFT-A proteins (*Figure 3E–G*, *Figure 3—figure supplement 1D*; *Video 3*). A pronounced fluorescence loss occurred predominately during the first 10–20 s of FLIP and some unbleached IFT-B proteins were present in the experimental bb-pool even in prolonged FLIP experiments (>30 s; *Figure 3C,D*, *Figure 3—figure supplement 1B*). For the NG-IFT54 bb-pool, this plateau was reached after ~15 s (STD 4.4 s, n = 14; *Figure 3—figure supplement 1F*) of FLIP illumination when the normalized signal strength of the experimental basal body was reduced to ~52% (STD 9.2%, n = 14; *Figure 3—figure supplement 1G*) of its initial level. After

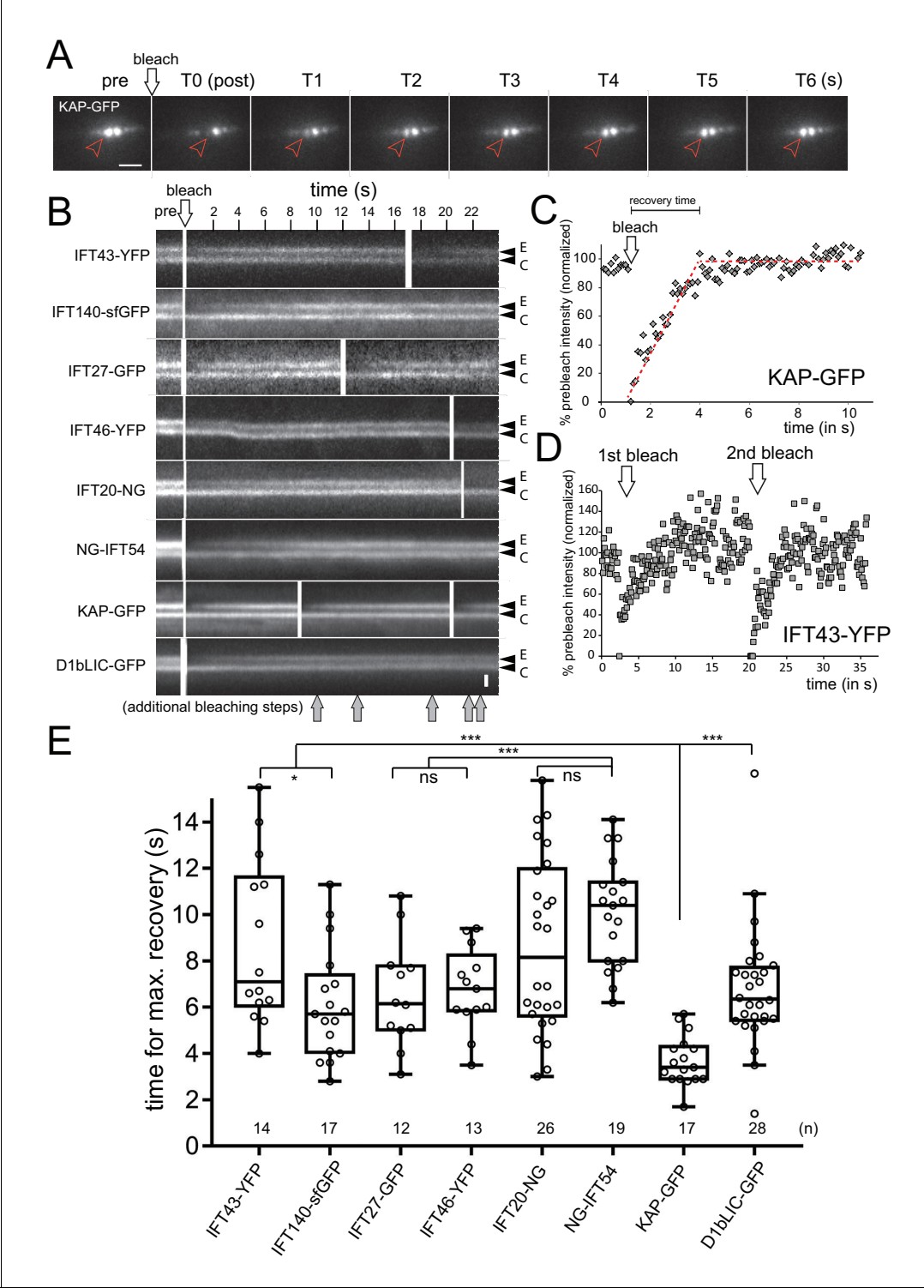

**Figure 2.** IFT proteins in the pool are exchanged at distinct rates. (A) Series of still images showing a cell expressing KAP-GFP before (pre) and immediately after bleaching of one of the two IFT bb-pools surrounding the flagellar basal bodies (arrowhead; T0), and during recovery (in s). Bar = 2 μm. (B) Kymograms from FRAP experiments after bleaching of one IFT bb-pool followed by signal recovery; repeated bleaching was used in some experiments (arrows, bottom). The experimental (E) and control (C) pool are indicated. Bar = 1 μm. (C,D) Quantitative analysis of FRAP for KAP-GFP (C) and IFT43-YFP (D) after bleaching of the experimental basal body; data are normalized for the unbleached basal body. The time needed for recovery was determined manually by determining the point of interception of lines along the slope during recovery and the plateau of the recovered signal (red

*Figure 2 continued on next page*

*Figure 2 continued*

dashed lines). (E) Box plot comparing FRAP times for the FP-tagged IFT proteins. The individual measurements, the median, and the quadrants are indicated. The number of measurements (n) and the result of a paired T-test are indicated. ns, not significant (p>0.05); *p≤0.05; **p≤0.01; ***p≤0.001.

The following figure supplement is available for figure 2:

**Figure supplement 1.** Rapid and local photobleaching using a focused laser beam.

switching off the FLIP laser, the basal body signal recovered and retrograde IFT recommenced (*Figure 3—figure supplement 1A,C*). We conclude that most of IFT27 and the tested IFT-A and motor subunits do not reenter the bb-pool whereas IFT20/46/54 re-enter the bb-pool after returning from the cilium.

## Anterograde trains are largely assembled from proteins freshly recruited to the basal bodies

The above experiments raise the question whether the IFT-B proteins returning via retrograde IFT into the bb-pool will be released with a delay into the cb-pool or reused directly in subsequent anterograde IFT trains without first cycling through the cb-pool. In the latter scenario, the signal strength of anterograde traffic should decrease progressively as the return of unbleached protein from the flagellum to the bb-pool is prevented. However, anterograde IFT trains containing unbleached IFT20/46/54 continued to enter the experimental flagella even after prolonged (>60s) FLIP illumination with a frequency comparable to anterograde IFT in the control flagellum (*Figure 3C*, *Figure 3—figure supplement 1A*). The signal strength of such trains, however, was generally below those in the control flagellum indicating that these trains contain both bleached proteins derived from retrograde trains and unbleached proteins recruited from the cell body. The signal strength of such trains could provide a measure of how much bleached and unbleached IFT-B protein was used for their assembly. A precise quantification is impeded because signal strength is affected by the absence of retrograde traffic and progressive loss of signal from stationary IFT in the experimental flagellum. To adjust for these shortcomings, we first analyzed the IFT-A protein IFT43-YFP, whose bb-pool signal is not affected in the FLIP assay (*Figure 3F,G*, *Figure 3—figure supplement 1D*). The signal strength of trains exiting the experimental bb-pool under FLIP illumination was ~10% below those in the control flagellum (n = 8, 4–7 IFT trains were measured in each cilium). A similar comparison for NG-IFT54 showed a 40% intensity decrease of trains in the experimental flagellum compared to trains in the control flagellum (n = 11; 4–8 measurements per cilium). The data indicate that a portion of the IFT-B protein NG-IFT54 in anterograde IFT trains is directly derived from retrograde IFT trains. With the qualification that IFT20/46/54 are partly salvaged, we conclude that *C. reinhardtii* possess a largely open IFT system, in which proteins are recruited from the cb-pool to the basal bodies, assembled into trains, and travel once through the flagellum before being disassembled and released back into the cell body pool.

When FLIP bleaching at one ciliary tip of a given cell was continued for several minutes the signal of IFT trains progressively diminished in both cilia (*Figure 3—figure supplement 2*). After ~6–9 min of FLIP, fluorescent IFT traffic diminished indicating that the entire cellular IFT-FP pool was near exhaustion. Sporadic residual NG-IFT54 traffic allowed for the observation of individual trains presumably containing only one or a few copies of unbleached NG-IFT54 (*Figure 3—figure supplement 2C*). Unbleached NG-IFT54 was evenly distributed between the

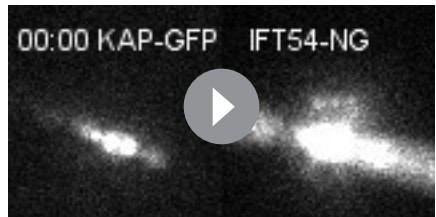

**Video 2.** FRAP analysis of the basal body-associated IFT pool. TIRF videos of cells expressing KAP-GFP (left) or the IFT particle protein NG-IFT54 (right) showing the IFT pools at the basal bodies. The pool facing to the right in each pair was bleached using brief illumination with a focused laser beam. Note fast recovery of the KAP-GFP signal and slower recovery of NG-IFT54. Images were acquired at 10 fps and playback is set at 20 fps (2 × speed). The timer counts mm:ss.

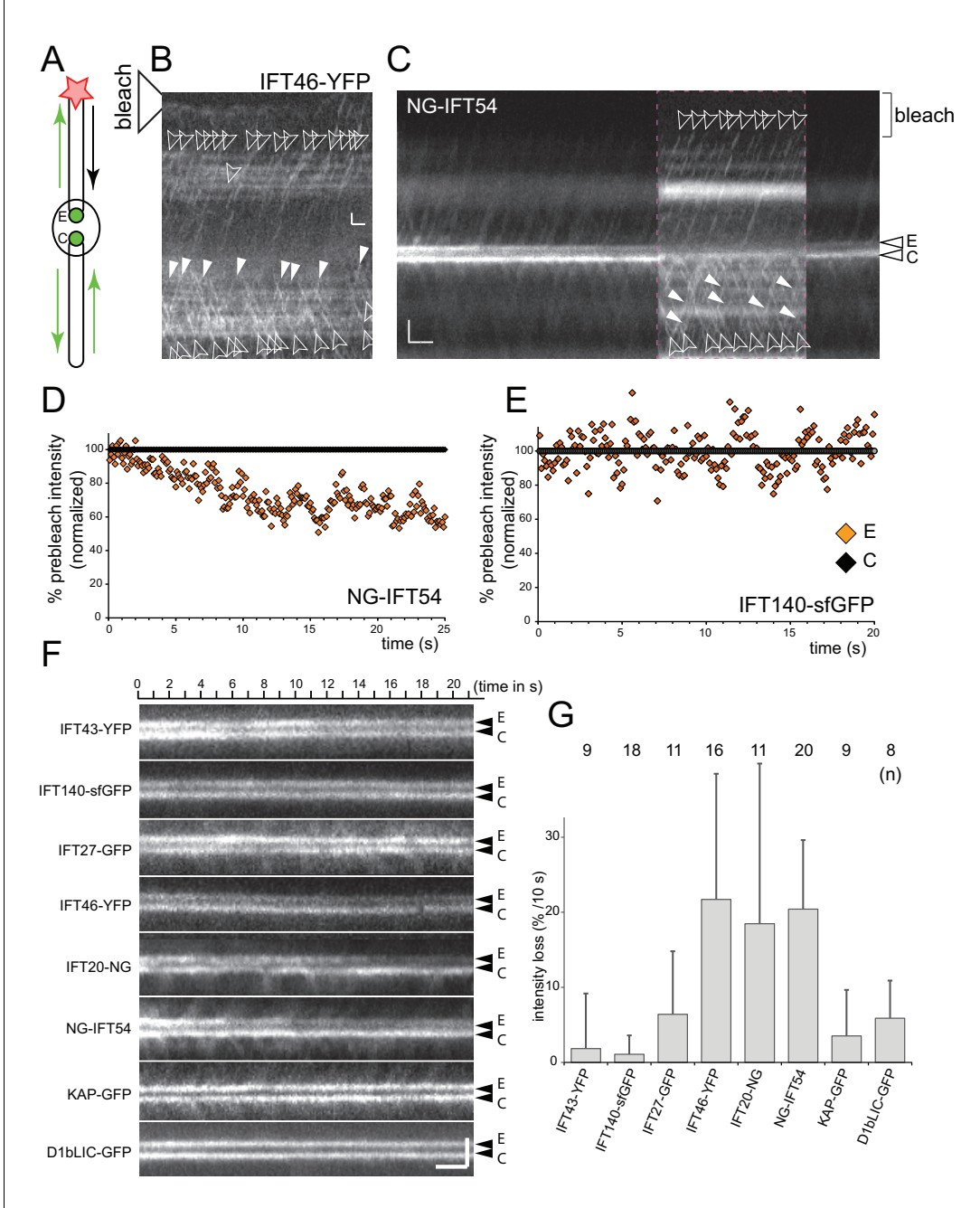

**Figure 3.** IFT-B, but not IFT-A or motor proteins, return to the basal body pool. (A) Schematic presentation of the FLIP assay to analyze the return of IFT proteins from the flagella to the bb-pool. The star indicates the laser beam positioned at the tip of the experimental (E) flagellum. (B) FLIP bleaching at the tip of the E flagellum prevented the return of unbleached IFT46-YFP via retrograde IFT; note retrograde tracks in the control (C) flagellum (full arrowheads). See *Figure 3—figure supplement 1A* for a more detailed analysis of IFT traffic under FLIP illumination. Bar = 1 μm 1 s. (C) Kymogram of a FLIP experiment of a NG-IFT54 cell. FLIP illumination decreased the signal of the E basal body. Unbleached anterograde trains (open arrowheads) continue to enter the E cilium in the absence of fluorescent retrograde traffic. To better visualize IFT traffic the focus level was changed during part of the recording (dashed square). Bar = 2 μm 2 s. (D,E) Quantitative analysis of FLIP of the E basal body of a NG-IFT54 and an IFT140-sfGFP cell; the data are normalized for the control (C) basal body. (F) Kymograms of FLIP experiments using the IFT-FP strains. The experimental (E; top) and control (C) pool are indicated. Bars = 2 μm 2 s. (G) Bar graph showing the average loss of signal of the E basal body during the first 10 s of FLIP illumination. The standard deviation and the number of cells analyzed are indicated. See *Figure 3—figure supplement 1E* for a box plot presentation of the data.

The following figure supplements are available for figure 3:

*Figure 3 continued on next page*

*Figure 3 continued*

**Figure supplement 1.** FLIP illumination prevents return of unbleached proteins to the cilary base.
**Figure supplement 2.** Bleaching of the entire NG-IFT54 protein pool.
**Figure supplement 3.** A portion of the pool consists of disassembling IFT-B complexes.

two flagella. In control cells imaged for similar periods of time without being directly hit by the FLIP laser, dense IFT traffic albeit of reduced intensity persisted (*Figure 3—figure supplement 2D*). The data suggest that NG-IFT54 returning to the cb-pool will eventually be reused and that bleached and unbleached proteins mix in the cb-pool and are stochastically recruited to the basal body pools.

## Half of the NG-IFT54 pool is derived from retrograde trains

The above data indicate that the bb-pool consists in parts of disassembling IFT-B complexes. The approach of bleaching trains only at the tip still allows unbleached retrograde trains *en route* and IFT proteins transiently stationary along the length of the cilium to return to the bb-pool. To determine the dynamics of the disassembling IFT-B bb-pool, we moved the focused laser beam from the flagellar tip to the base bleaching almost the entire NG-IFT54 population in the experimental flagellum (*Figure 3—figure supplement 3A*). This approach offers a clear onset and an extended period during which only bleached trains return to the bb-pool. After the bleaching step, the signal strength of the experimental bb-pool declined by 52% (STD 8.5%, n = 6) over a period of 5.8 s (STD 0.9 s, n = 6) before reaching a plateau (*Figure 3—figure supplement 3B–D*). Thus, ~50% of the NG-IFT54 in the bb-pool is derived from retrograde trains. The first unbleached retrograde train returned after 10.9 s (STD 1.8 s, n = 22) to the experimental basal body (*Figure 3—figure supplement 3B,C*). Concomitantly with the return of unbleached NG-IFT54 via retrograde traffic, the bb-pool recovered in strength on average in 16.6 s after the bleaching step (STD 2.4s, n = 6) to levels of the control basal body (*Figure 3—figure supplement 3B–D*). The data suggest that NG-IFT54 entering the bb-pool by retrograde traffic remains for ~6 s in the pool before its release into the cell body or reuse in anterograde trains. In similar experiments using KAP-GFP or the IFT-A protein IFT43-YFP, the signal strength of the bb-pool attached to the experimental flagellum remained essentially unaltered indicating that these proteins are released into the cb-pool upon return from the cilium (n = 4 and 5, respectively; *Figure 3—figure supplement 3E–H*).

## Mammalian IMCD3 cells and *Tetrahymena thermophila* have open IFT systems

In contrast to our data on *C. reinhardtii*, observations made in *Trypanosoma brucei* indicated a closed IFT system in which proteins move back and forth between the bb-pool and flagellum with negligible exchange with the cb-pool during the duration of the experiment (*Buisson et al., 2013*). To determine the prevalence of open and closed systems, primary cilia of IMCD cells expressing IFT88-YFP and motile cilia of the ciliate *Tetrahymena thermophila* expressing GFP-Dyf-1 (IFT70) were analyzed (*Figure 4*).

In primary cilia of IMCD cells, IFT88-YFP moved in typical anterograde IFT trains but was

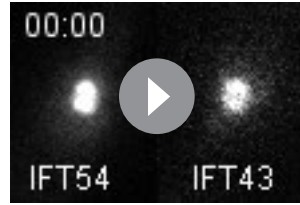

**Video 3.** FLIP analysis of the basal body-associated IFT pool. Shown are the IFT pools for NG-IFT54 (left) and IFT43-YFP (right) during FLIP illumination. The tip of the flagellum attached to the upper basal body in each pair was illuminated with a blinking laser beam to prevent the return of unbleached IFT proteins via retrograde IFT to the attached basal body. The flagella are out of focus and the flashing laser beam is out of the field of view. Note progressive loss of fluorescence for NG-IFT54 in the experimental bb-pool until it reaches a plateau; IFT43-YFP levels in the experimental bb-pool are not affected by FLIP illumination when compared to the control pool but a general loss of signal strength occurs due to the low photostability of YFP and the prolonged illumination. Images were acquired at 10 fps and playback is set at 30 fps (3 × speed). The timer counts mm:ss.

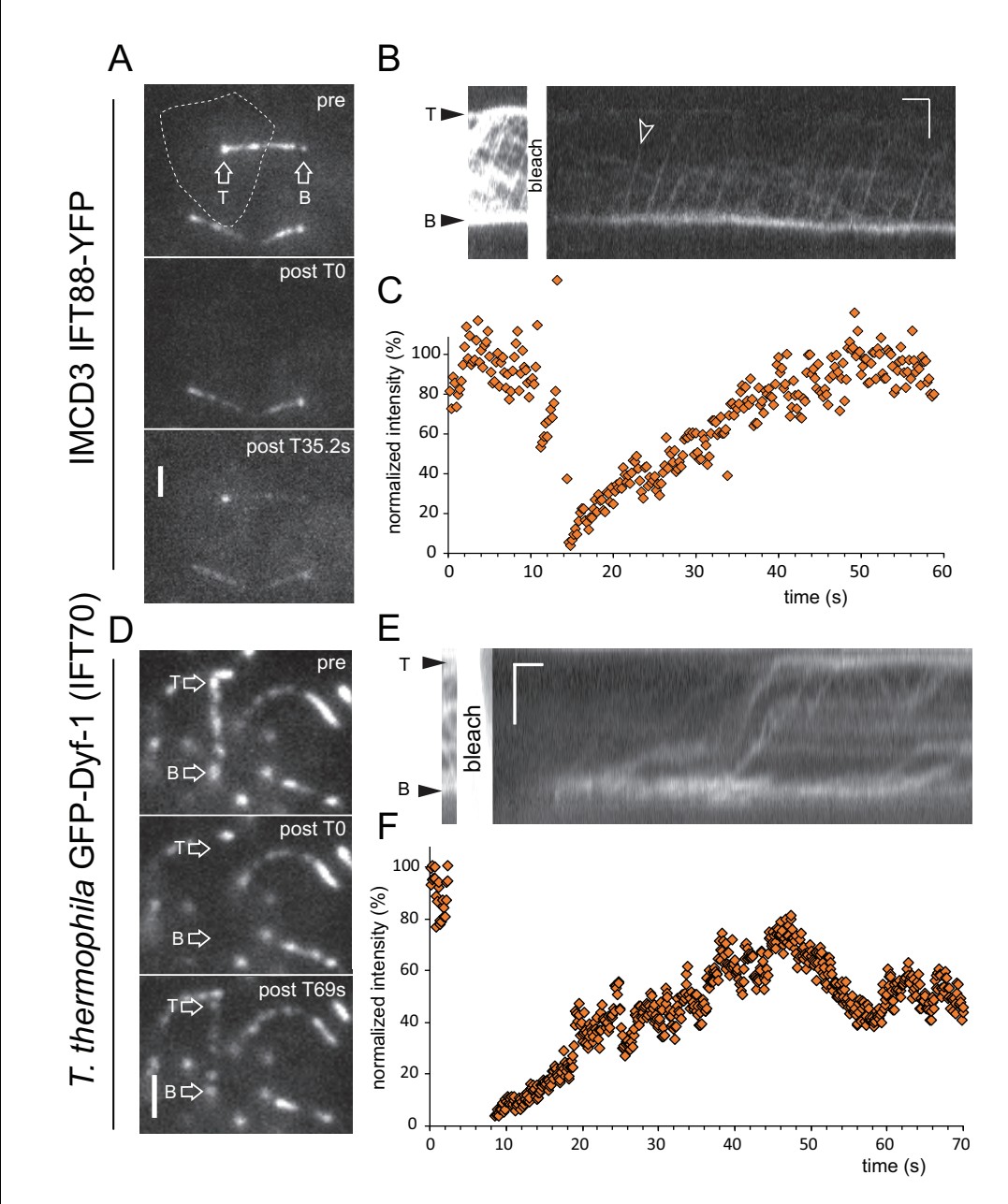

**Figure 4.** IMCD cells and *T. thermophila* have open IFT systems. (**A**) Still images of an IMCD cell expressing IFT88-YFP before (pre), immediately after photobleaching of the entire cilium plus basal body (post), and during recovery. The dotted line marks the cell bounderies. Bar = 1 µm. (**B**) Corresponding kymogram (Bar = 2 µm 5s) and (**C**) quantitative analysis of the signal at the basal body. (**D**) Still images showing a detail of the cortex of a *T. thermophila* cell expressing GFP-Dyf-1 before (pre), immediately after photobleaching of the entire cilium plus basal body (post), and during recovery. Bar = 1 µm. (**E**) Corresponding kymogram (Bar = 2 µm 5s) and (**F**) quantitative analysis of the signal at the basal body.

also present in stationary aggregates resulting in a high background (*Figure 4A,B*). After photo-bleaching of the entire cilium including its bb-pool, the basal body signal recovered to 60–95% within 16.8 s (STD 7.2 s, n = 11, normalized using the cilium of a neighboring unbleached cell; *Figure 4A,C*); the first unbleached IFT trains entered the cilia after an average of 18.1 s (STD 9.5 s, n = 39, *Figure 4B*;

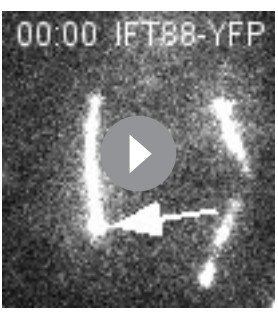

**Video 4.** FRAP analysis of IFT88-YFP in primary cilia of IMCD3 cells. The ciliary base of the photobleached cilium is marked by an arrow. Note that the recovered signal strength of the experimental cell is similar to those of the unbleached control cells. Still images of this movie are shown in *Figure 4A*. Images were acquired at 5 fps and playback is set at 40 fps (8 × speed). The timer counts mm:ss.

*Video 4*). The data reveal that IFT88-YFP in IMCD cells is continuously recruited from the cell body to the basal bodies and assembled into IFT trains.

*T. thermophila* possesses ~800 cilia; these mobile cells were immobilized by compression between the cover glass and slide allowing us to image IFT as visualized by the expression of GFP-Dyf-1 (IFT70) in a null background (*Dave et al., 2009*; *Jiang et al., 2015*). After bleaching of a single cilium with its bb-pool, the recovery of the basal body signal became quickly apparent and reached a maximum of 74% (STD 25%, n = 10) of the prebleach value in 29 s (STD 8s, n = 10, *Figure 4D–F*); unbleached anterograde trains were observed after 17.1 s (STD 8 s, n = 20; *Video 5*) and the ciliary signal recovered partially indicating a continuous recruitment of cell body GFP-Dyf-1 to the basal bodies for assembly into IFT trains. In summary, FRAP analysis indicates open and semi-open IFT systems in all three cell types.

## IFT trains exit the pool from distinct sites

To determine the size of the IFT bb-pool in *C. reinhardtii*, we focused on NG-IFT54 and KAP-GFP, which due to their distal position on the basal body often allowed for simultaneous imaging of the bb-pool and departing trains (*Figure 5*). The departure of an IFT train resulted in a transient reduction of signal strength at the bb-pool of ~16.1% (STD 7.2%, n = 18) for KAP-GFP, and 12.7% (STD 5.8, n = 20) for NG-IFT54 (*Figure 5A–C*). Similarly, the arrival of a train increased the NG-IFT54 bb-pool signal by up to 25% (*Figure 5C*). In this analysis we focused on bright trains allowing for a clear correlation between train trajectories and fluorescence loss in the bb-pool. Bright trains alternated with dimmer trains (*Ludington et al., 2013*), which changed the bb-pool signal by an estimated 5–10%. The data reveal that the departure (or arrival) of a single IFT train appreciably affects the strength of the bb-pool suggesting that the amount of IFT protein in the bb-pool is equivalent to a number of IFT trains in the upper single or lower double digits. NG-IFT54 trains, which could be best studied due to their bright and stable signal, exited the bb-pool at distinct sites resulting in a local depletion of the bb-pool (*Figure 5D*, *Video 6*). Kymograms obtained by scanning along two or three distinct lines parallel to the basal body-flagellum axis of a given cell were merged to visualize the exit of trains from distinct sites of the bb-pool and their journey along the flagellum (*Figure 5E–I*). While no particular pattern was observed, the data nevertheless indicate that consecutive trains are often released from distinct sites of the bb-pool.

## IFT proteins are recruited and assembled sequentially into anterograde trains

To determine the time required for the assembly of an anterograde IFT train, we measured the interval between the bleaching of one bb-pool and the departure of the first unbleached train into the attached flagellum (*Figure 6A*). The untreated basal body and its flagellum served as a crucial control because exceedingly strong or

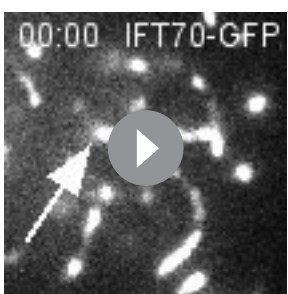

**Video 5.** FRAP analysis of GFP-Dyf-1 (IFT70) in *T. thermophila* cilia. The basal body of the bleached cilium is marked by an arrow. For clarity, most of the over-exposed frames of the bleaching step were deleted. Still images of this movie are shown in *Figure 4D*. Images were acquired at 10 fps and playback is set at 80 fps (8 × speed). The timer counts mm:ss.

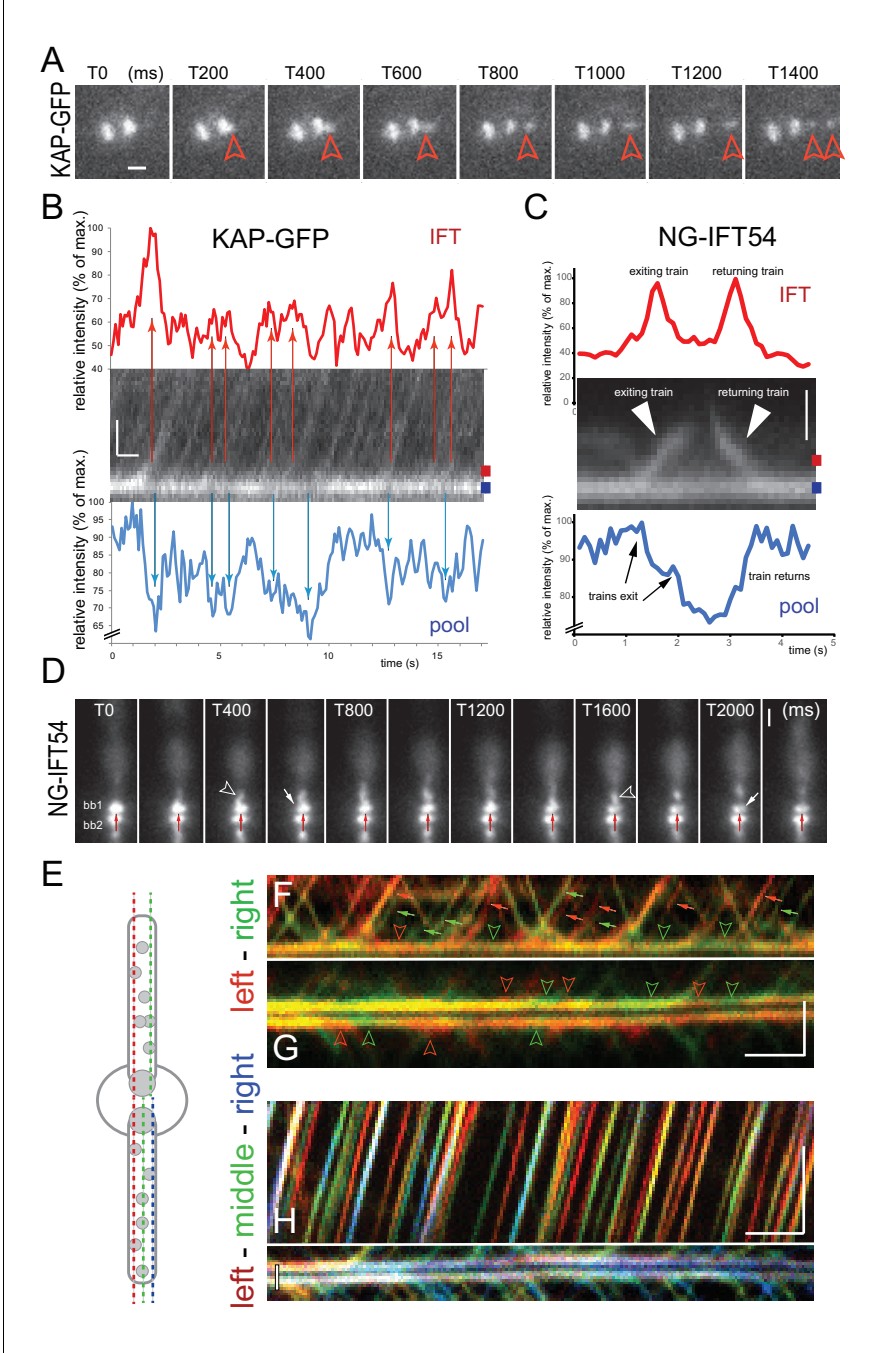

**Figure 5.** The basal body pool contains IFT proteins equivalent to several IFT trains. (**A**) Series of still images from a cell expressing KAP-GFP showing the departure of an IFT train (red arrowhead). The intensity of the bb-pool decreases concomitantly with the departure of the train. Bar = 1 μm; the time points in ms are indicated. (**B**) Kymogram showing the bb-pool and proximal regions of one flagellum of a cell expressing KAP-GFP. Intensity profiles (in percent of the maximum intensity) for the bb-pool (blue) and flagellum (red) depict the correlation between the exit of IFT trains (red arrows) and a transient decrease of fluorescence in the pool (blue arrows); the positions of the lines used to generate the intensity profiles are indicated. Bars = 1 s 1 μm. (**C**) Kymogram and its analysis as described in B but for NG-IFT54. Bar = 1 μm. (**D**) Series of still images from a cell expressing NG-IFT54 showing two IFT trains departing from the upper basal body (bb1; arrowheads). The first train departs from the left (T400) side of the bb-pool while the second train exits the pool at the right side causing a local loss of fluorescence in the pool. The red arrow near bb2 serves as a fiduciary marker. Bar = 1 μm. (**E**) Schematic presentation of the position of the lines used to generate the kymograms shown in **F–I**. (**F,G**) Merged kymograms

*Figure 5 continued on next page*

*Figure 5 continued*

showing the two sides of the bb-pool and cilium in red and green. Note the alternation of predominantly red and green trains (small arrows) and red and green areas (open arrowheads) in the bb-pool indicative for a transient reduction of NG-IFT54 in one side of the pool. Bars = 2 s 2 μm. (**H,I**) as in **F** and **G**, but showing the left, middle, and right positions of the pool and cilium in red, green, and blue respectively. Bars = 2 s 2 μm.

long laser pluses disturbed IFT traffic in both flagella (not shown). Typically, bleaching of the bb-pool was followed by an interruption in the traffic of fluorescent anterograde IFT in the experimental flagellum; we refer to this period as the gap. After the gap anterograde traffic recommenced with prebleach or almost prebleach intensity and frequency (*Figure 6B*). Infrequent 'early bird' IFT trains could result from incomplete bleaching of the bb-pool; a limited replacement of bleached protein in the assembling trains with unbleached proteins from the surroundings is also possible. Occasionally, we also observed retrograde trains apparently making a U-turn near the ciliary base (not shown).

The presence of a distinct gap after bleaching of the bb-pool suggests that IFT trains containing the bleached proteins exit the bb-pool before new trains assembled from unbleached proteins emerge. The duration of the gap varied considerably for the different IFT proteins analyzed and ranged from ~7 s for IFT140-sfGFP to ~2.5 s for D1bLIC-GFP (*Figure 6C*; *Video 7*). The differences in the duration of the gap between each of the two IFT-A, -B1, and -B2 proteins as well as those between the IFT-A and -B2 proteins were not significant while those between IFT-A and -B1 and the motors were. The observation suggests that distinct IFT proteins need different time spans to transition through the bb-pool from recruitment to release via anterograde IFT.

To substantiate this observation, we generated a strain expressing KAP-GFP and IFT140-mCherry in the corresponding *fla3 ift140* double mutant background (*Figure 7A*). We then analyzed the duration of the gap between bleaching of the bb-pool and release of the first unbleached KAP-GFP and IFT140-mC trains into the attached flagellum (*Figure 7B*). In most experiments (42 of 49), trains labeled only with KAP-GFP reappeared before trains marked by both KAP-GFP and IFT140-mCherry; in the remaining experiments both markers were present in the first postbleach train (*Video 8*). In the two-color experiments, the average duration of the gaps between bleaching of the bb-pool and the discharge of the first unbleached KAP or IFT140 trains were similar to those determined in the single-tag strains (*Figure 7C*). The distinct properties of the two proteins were also apparent in two-color FRAP analysis of the bb-pool: The KAP-GFP signal was restored significantly before that of IFT140-mCherry (n = 3, *Figure 7D,E*). The data indicate that IFT140-sfGFP is recruited early during the assembly of a given IFT train while KAP-GFP is added later (*Figure 7—figure supplement 1*). In a similar analysis using a strain expressing NG-IFT54 (IFT-B1) and IFT140-mCherry (IFT-A), the post-bleach trains mostly contained both fluorescent IFT-proteins (13 of 15 experiments; in two experiments the first train contained NG-IFT54 but not IFT140-mC; not shown). We propose that IFT trains are assembled by sequential addition of distinct IFT proteins/subcomplexes and that trains in different states of assembly are lined up in the bb-pool to be successively dispatched into the flagellum upon completion.

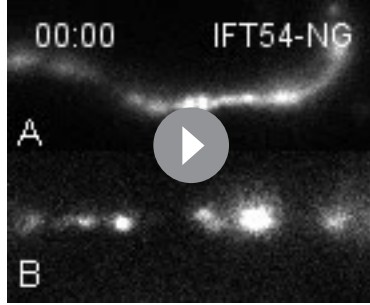

**Video 6.** Spatial analysis of IFT proteins in the pool and flagellum. IFT in NG-IFT54 cells was monitored at the focus level of the basal bodies (top) or of the flagellum (bottom). Top) The two basal body-associated IFT pools are positioned in the center. Note the fluctuation in the signal intensity in the pools as trains exit and arrive. Often, only a part of the pool signal is bleached indicative for the exit of trains from a particular region of the pool. Bottom) IFT trains moving inside a flagellum. Some trains move closer to the top, other closer to the bottom of the flagellum. Images were acquired at 10 fps and the video is displayed at 20 fps (2 × speed). The timer counts mm:ss.

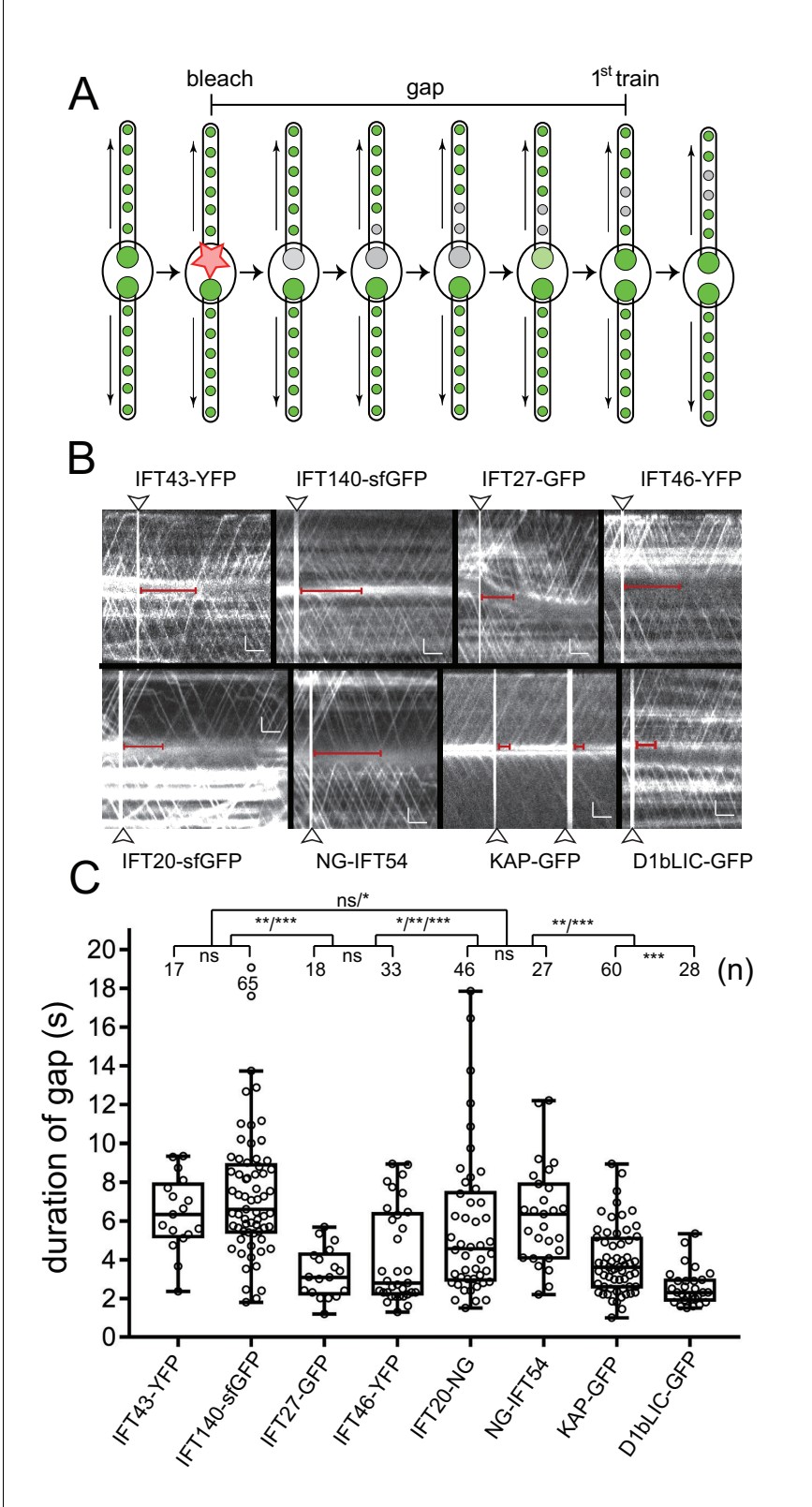

**Figure 6.** Bleaching of the basal body pool is followed by a gap in IFT traffic. (**A**) Schematic presentation of the gap assay to determine the interval between bleaching of the bb-pool (bleach, indicated by the star) and recommencement of fluorescent anterograde IFT traffic (first train). (**B**) Representative gap analysis for the eight FP-tagged IFT proteins. Arrowheads mark the bleaching steps; the red bars indicate the gap between the bleach

*Figure 6 continued on next page*

*Figure 6 continued*

and the dispatch of the first unbleached IFT train. Bars = 2 s 2 μm. (**C**) Quantitative analysis of the gap data. The standard deviation, number of gaps analyzed, and the results of a paired T-test are indicated. ns, not significant (p>0.05); *p≤0.05; **p≤0.01; ***p≤0.001.

## Tubulin binds briefly before the departure of IFT trains into the cilium

IFT trains transport axonemal proteins into flagella but it remains unclear when and where cargo proteins attach to the trains (*Craft et al., 2015*; *Qin et al., 2004*; *Wren et al., 2013*). Axonemal cargoes could already bind to IFT complexes in the cb-pool (e.g., on post-Golgi vesicles) or later during train assembly in the bb-pool (*Wood and Rosenbaum, 2014*). The above described approach of determining the duration of the gap in anterograde traffic after bleaching of the bb-pool has the potential to discriminate between these possibilities: If a cargo binds to IFT already prior to the recruitment to the bb-pool the gap should be in the range of that determined for the IFT proteins; shorter gaps would indicate loading at a later stage of train assembly. A prerequisite for this approach is a dense and regular cargo traffic as it has been reported for sfGFP-tubulin, which during flagellar regeneration enters cilia with an anterograde frequency of ~20/minute and even higher frequencies during early regeneration (*Craft et al., 2015*). Cells expressing sfGFP-α-tubulin at ~10% of the endogenous α-tubulin were first deflagellated by a pH shock and allowed to initiate flagellar regeneration for >20 min. sfGFP-tubulin already incorporated into the flagella was photobleached until IFT traffic of sfGFP-tubulin became visible (*Figure 8A*). Then, one basal body region was photobleached using the microtubular roots to position the beam (*Figure 8B*). sfGFP-tubulin traffic into the attached flagellum resumed almost immediately (*Figure 8C,D*). In detail, the gap between the bleach and the first sfGFP-tubulin trajectory was 1.9 s (STD 1.7 s, n = 25) for the experimental flagellum and 2.1 s (STD 1.6 s, n = 25) for the control flagellum; the difference was not significant (2-tailed t-test p=0.74). To ensure proper bleaching of sfGFP-tubulin near the flagellar base, we used an extended laser beam with a diameter of ~2 μm. sfGFP-tubulin traffic into the flagella resumed after on average 2.5 s (STD 1.5 s, n = 22). Notably, a similar short gap was also observed for D1bLIC, a subunit of IFT dynein, which is transported as a cargo on anterograde trains. Based on these data, we propose that sfGFP-tubulin and IFT dynein associate to IFT trains at the flagellar base and briefly before departure into the cilium.

## Discussion

We used in vivo imaging in *C. reinhardtii* to study the assembly of IFT trains. IFT trains were first observed using DIC microscopy as birefringent particles moving up and down the flagella (*Kozminski et al., 1993*). Using GFP-tagged proteins, discrete IFT trains traveling inside cilia were observed in various organisms (*Besschetnova et al., 2009*; *Mueller et al., 2005*; *Orozco et al., 1999*; *Williams et al., 2014*). Correlative light-electron microscopy identified IFT trains at as electron opaque arrays with a repetitive ultrastructural moving between the ciliary membrane and the axoneme (*Kozminski et al., 1995*; *Stepanek and Pigino, 2016*). The pool of IFT proteins surrounding the basal bodies apparently lacks the well-defined

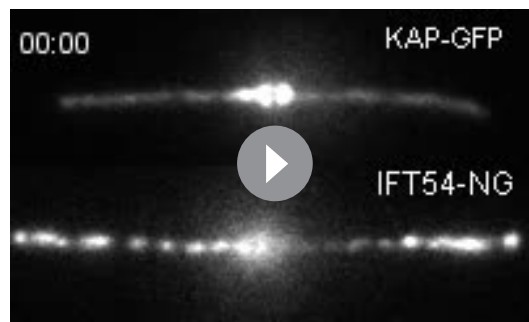

**Video 7.** Photobleaching of the IFT pool is followed by a gap in anterograde IFT of fluorescent protein. One of the two basal bodies of a KAP-GFP (A, top) and a NG-IFT54 cell (B, bottom) was bleached. This results in an interruption of anterograde IFT of fluorescent protein. The gap is short for KAP-GFP and longer for NG-IFT54. This can be best viewed by first focusing on the KAP-GFP flagellum pointing to the right. Once the first unbleached KAP-GFP trains re-appear after the bleaching step, the viewer should focus on the right NG-IFT54 flagellum: At this time point only retrograde but no fluorescent anterograde trains are visible in the NG-IFT54 flagellum. Anterograde IFT of NG-IFT54 will resume a few seconds later. Images were acquired at 10 fps and the video is displayed at 20 fps (2 × speed). The timer counts mm:ss.

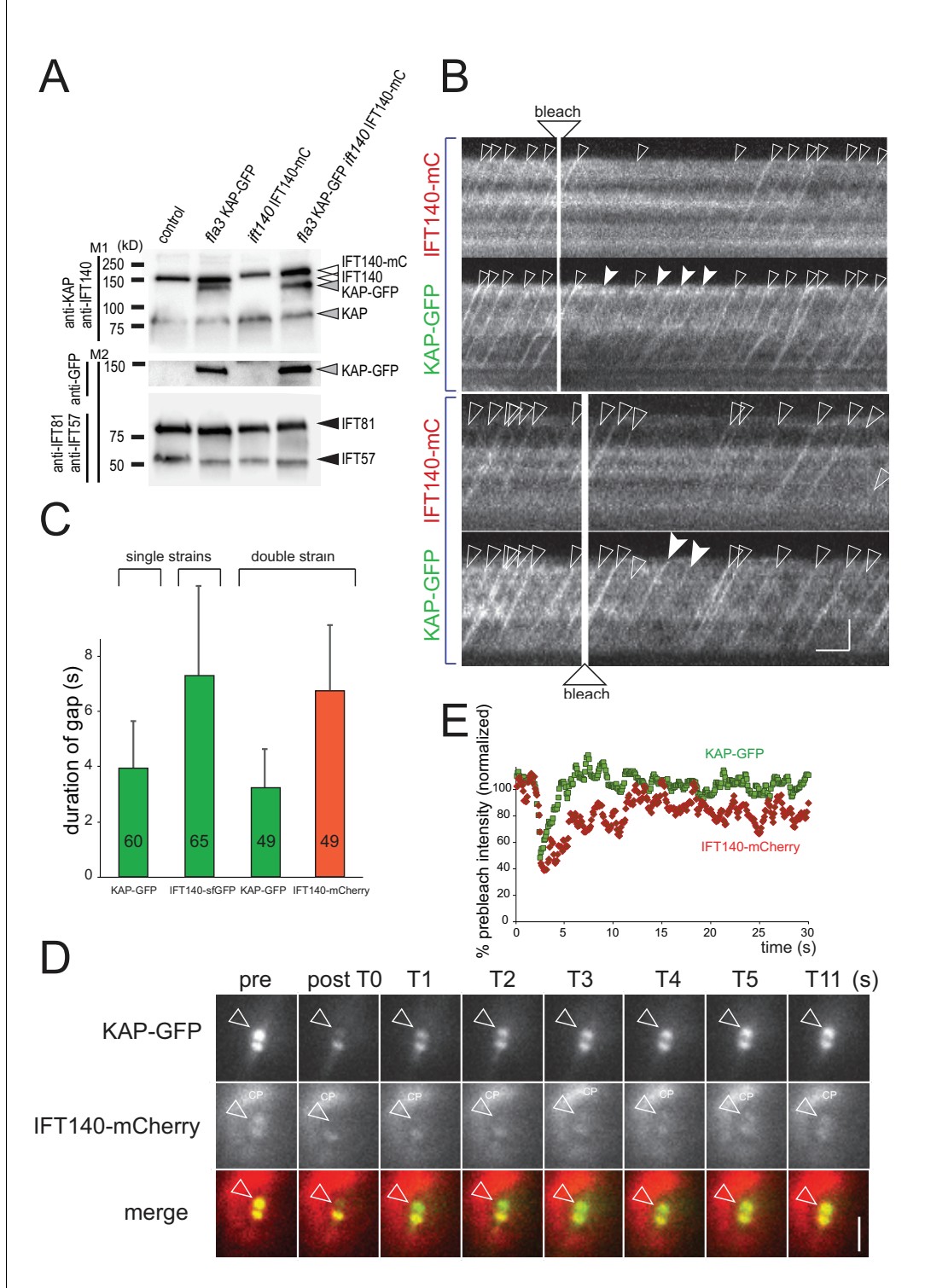

**Figure 7.** IFT140 is recruited early, KAP is recruited late during IFT train assembly. (**A**) Western blot analysis of the flagellar 'membrane + matrix' fraction of a control (G1), the KAP-GFP, the IFT140-mC, and the KAP-GFP IFT140-mC strains; note that KAP-GFP was expressed in the *fla3* which has a mutated endogenous KAP of reduced function. Two membranes (M1 and M2) loaded with the same amount of sample were stained with a mix of anti-KAP and anti-IFT140 (M1) or of IFT81 and IFT57 as loading controls (M2), the latter membrane was subsequently stained with anti-GFP. Molecular weight markers are indicated (in kD). (**B**) Two examples showing the recommencement of IFT traffic after photobleaching of the basal body in cells expressing KAP-GFP and IFT140-mCherry. Trains containing only unbleached KAP-GFP (filled arrowheads) appeared before trains containing both unbleached IFT140-mCherry and unbleached KAP-GFP (open arrowheads). Bars = 2 s 2 μm. (**C**) Histogram comparing the duration of the gap after

*Figure 7 continued on next page*

*Figure 7 continued*

bleaching of the bb-pool for strains expressing KAP-GFP, IFT140-sfGFP, or KAP-GFP and IFT140-mCherry. (D) FRAP analysis of the bb-pool in a cell expressing KAP-GFP and IFT140-mCherry. Note that the KAP-GFP signal recovered before the IFT140-mCherry signal. CP, chloroplast. (E) Quantification of the KAP-GFP and the IFT140-mCherry signals of the experiment depicted in C. Data are normalized for the signal strength of the control basal body pool.

The following figure supplement is available for figure 7:

**Figure supplement 1.** Model of the temporal and spatial organization of IFT proteins in the basal body pool.

ultrastructure of the trains moving inside cilia (*Rogowski et al., 2013*). Based on the brightness of signals representing immunostained or FP-tagged IFT proteins, the bb-pool is several times the size of a single train. Due to this crowded situation, imaging of individual IFT trains from their inception to release into the cilium has not been achieved. Here, we used in vivo imaging and photobleaching to study the assembly of IFT trains. Our two main conclusions are that the IFT system is largely open with proteins constantly exchanging between the basal body-flagellum compartment and the cell body and that the bb-pool consists of a queue of IFT trains in different stages of construction.

## The IFT system is open

IFT trains perpetually enter and exit the flagellum but the source of IFT proteins in the bb-pool and the fate of the IFT proteins returning to the pool are less clear. Two not mutually exclusive models have been proposed: A closed IFT system in which retrograde trains are remodeled in the bb-pool and reused in anterograde traffic and an open IFT system in which proteins are recruited to the bb-pool, move as trains through the cilium, and disperse back into the cb-pool (*Buisson et al., 2013*). Photobleaching of FP-tagged IFT proteins allows us to distinguish between these models. In a closed system, fluorescent anterograde traffic should quickly cease when the return of unbleached proteins to the flagellar base is prevented by photobleaching. In contrast, fluorescent anterograde traffic will continue in an open system until the entire cellular pool of the protein in question is bleached. Our data in *C. reinhardtii* revealed a largely open system for the IFT-B protein IFT27-GFP and the tested IFT-A and motor subunits and semi-open systems for the IFT-B proteins IFT20/46/54, where portions of the retrograde trains are reused in the assembly of anterograde trains (*Figure 8E*). FRAP analysis of one IFT-B protein each indicates that open or semi-open IFT systems are also in place in *T. thermophila* and IMCD cells.

In contrast, a closed IFT system was proposed for the IFT-B protein IFT52 in *T. brucei* (*Buisson et al., 2013*). Similar to our observations on *C. reinhardtii* IFT-B proteins, bleaching of the IFT52-GFP bb-pool in *T. brucei* was followed by a gap in anterograde traffic and IFT52-GFP from retrograde trains reentered the bb-pool. However, when anterograde IFT resumed the signal intensity of the trains was markedly below that of trains prior to the bleaching step and the total recovery of the bb-pool reached only ~50% of the prebleach intensity suggesting that the bleached proteins continue to cycle within the flagellum-basal body domain while an exchange with the cb-pool is negligible within the time span of the experiment. Possible reasons for the disparities between *C. reinhardtii*

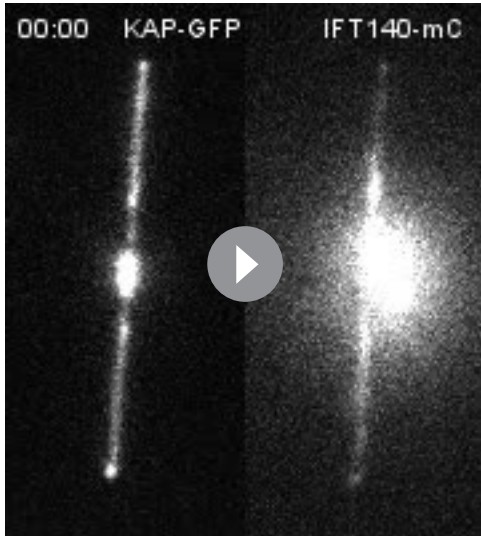

**Video 8.** Gap analysis of a cell expressing KAP-GFP and IFT140-mCherry. Follow the viewing instruction described for *Video 7*. Note brief (~2 s) gap between the first post-bleach KAP-GFP train and the first postbleach IFT140-mC train. Images were acquired at 10 fps and the video is displayed at 20 fps (2 × speed). The timer counts mm:ss.

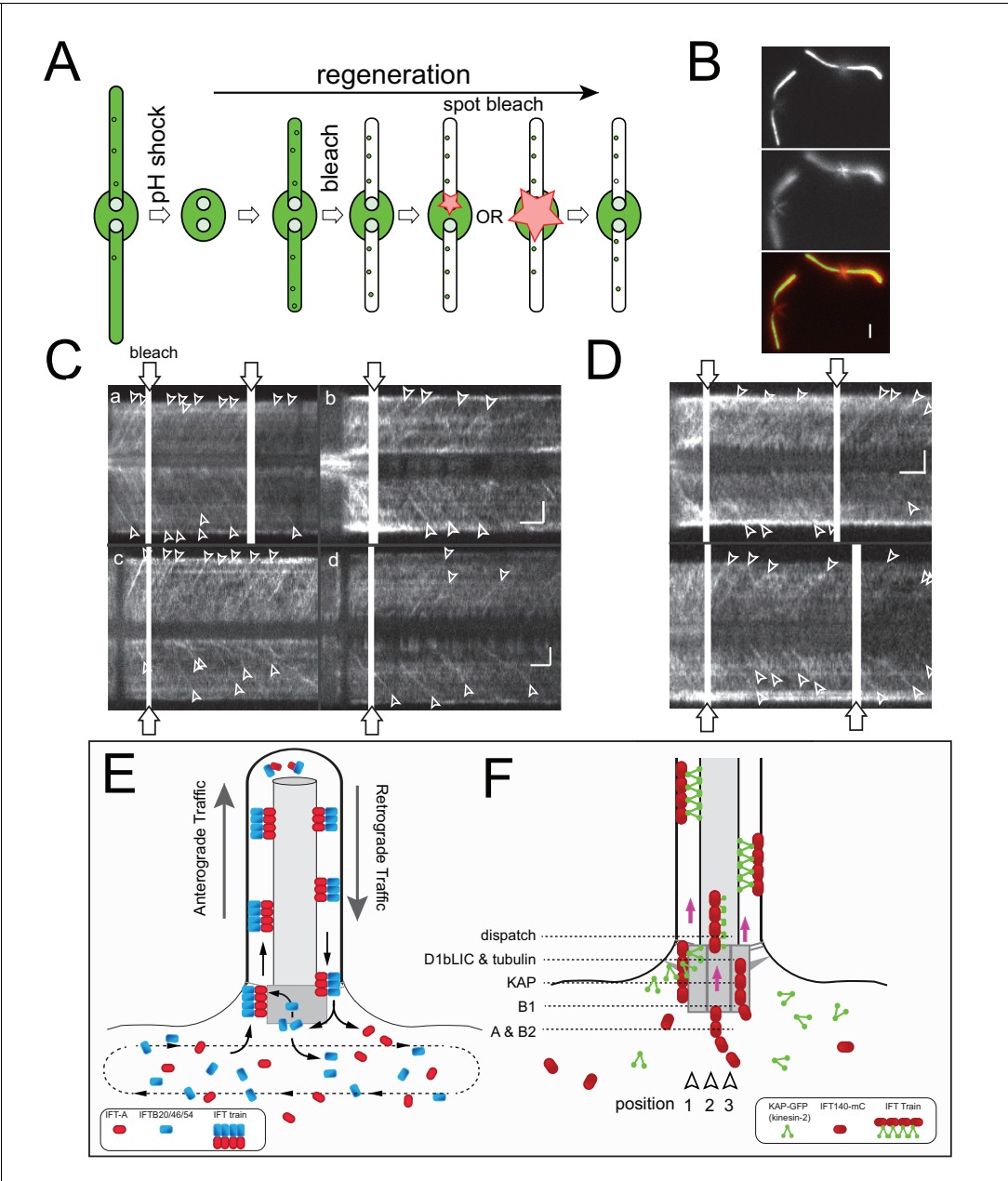

**Figure 8.** Tubulin is loaded briefly before IFT trains enter the cilium. (**A**) Schematic presentation of the experimental design used to determine the gap in anterograde traffic of sfGFP-tubulin after photobleaching of the basal body. Cells were deflagellated by a pH shock, allowed to initiate regeneration, and mounted for TIRF analysis. Then, the two flagella were partially bleached, followed by spot bleaching of either one or both basal body pools (for the latter the size of the laser beam was extended to ~2 µm), and TIRF analysis of tubulin traffic. (**B**) Two images from a focal series showing the flagella and the microtubular roots; the latter were used to position the cells with respect to the laser beam. The two images are displayed in red and green in the merged image. Bar = 2 µm. (**C**) Gallery of kymograms from experiments in which one of the two basal body pools was bleached. The bleached basal bodies are oriented to the top. Bars = 2 s 2 µm (in d for a, c, d; in b for b). (**D**) Gallery of kymograms in which both basal body pools were bleached. In **C** and **D**, trajectories indicating IFT of tubulin are indicated by arrowheads. Bars = 2 s 2 µm. (**E**) Schematic presentation of the open IFT system. IFT-A proteins (and IFT27 and the motors) are recruited to the pool, cycle once through the flagellum, and return into the cell body pool. IFT-B proteins return to the bb-pool to be either reused in subsequent IFT trains or released into the cell body pool. IFT proteins mix in the cb-pool and are randomly re-deployed into IFT trains. (**F**) Model of the temporal-spatial organization of IFT proteins in the basal body pool. We propose that IFT trains assemble in distinct positions near the basal body from which they are sequentially released into the cilium upon completion. Assembly commences with the arrival of IFT-A complexes which then combine with the other IFT-subcomplexes and motors as they move toward the distal end of the basal body. For simplicity, only IFT-A and kinesin-2 are depicted. See also *Figure 7—figure supplement 1*.

and *T. brucei* range from differences in the experimental set-up to fundamental differences in IFT. For example, in a semi-open system as described for the IFT-B proteins, the ratio between new recruitment from the cb-pool and reusage of proteins derived from retrograde trains could be different between species with *T. brucei* favoring reusage. To study IFT52 dynamics in *T. brucei*, the bb-pool was photobleached. This technique cannot distinguish whether the unbleached proteins in postbleach trains are derived from retrograde trains or recruited from the cb-pool. In contrast, the observation of stable fluorescent IFT traffic after prolonged bleaching of retrograde IFT clearly indicates that unbleached proteins are continuously recruited from the cell body for incorporation into IFT trains. Single molecule imaging showed that kinesin-2 and IFT dynein returning from cilia in *C. elegans* diffuse briefly near the base to turn around and re-enter the cilium supporting a closed IFT system (*Mijalkovic et al., 2017*; *Prevo et al., 2015*). While our data do not exclude that some motors remain in the basal body-cilium compartment for several rounds of IFT, the bulk of the motor subunits KAP-GFP and D1bLIC-GFP in anterograde trains of *C. reinhardtii* is recruited from the cb-pool (see *Figure 3—figure supplement 3G,H* for KAP-GFP). In their mature state, *C. elegans* cilia possess altered basal bodies and lack the transitional fibers (TFs), two structures that are likely to be critical for the recruitment of IFT proteins from the cb-pool to the ciliary base; these changes in could favor a more closed IFT system (*Nechipurenko et al., 2017*). Future studies using a range of imaging approaches and organisms are needed to evaluate the prevalence of closed and open IFT systems.

## Temporal and spatial organization of the IFT pool

Comparison of the in vivo signals of eight FP-tagged proteins revealed differences with respect to their distribution along the basal body axis and their distance to the basal bodies confirming previous observations based on antibody staining (*Brown et al., 2015*; *Hou et al., 2007*; *Richey and Qin, 2012*). It appears that different IFT proteins occupy distinct territories indicating that the bb-pool contains IFT subcomplexes or individual proteins rather than entire IFT particles. Super-resolution imaging is likely to provide a clearer image of IFT protein territories within the bb-pool. Our data also indicate a functional compartmentalization of the bb-pool into assembling anterograde trains and disassembling IFT-B complexes.

The FRAP pattern of anterograde traffic after bleaching of the IFT bb-pool further informs on the assembly of IFT trains. Bleaching of the bb-pool is typically followed by a marked gap in IFT traffic during which no visible or only a few dim anterograde IFT trains enter the flagellum. A distinct gap in IFT traffic was also observed for IFT52-GFP in *T. brucei* (*Buisson et al., 2013*). The presence of a gap indicates that bleached trains exit the bb-pool ahead of trains assembled later from unbleached proteins. Based on these data, we propose that the IFT trains queue in the bb-pool to be released in order with newly assembling trains added to the end of the queue.

The length of the postbleach gap varied between the IFT proteins analyzed. A short gap, as it was observed for the KAP and D1bLIC, indicates recruitment of a given subunit shortly before the train exits the bb-pool; conversely, a long gap as observed for the IFT-A and IFT-B2 subunits, indicates that a protein was recruited early during train assembly. The data imply that the bb-pool should contain more IFT-A proteins (in equivalents of IFT trains) than KAP. The use of different FPs, the distinct spatial distribution of IFT protein in the bb-pool, and the expected small scale of the expected differences prevented us from comparing the amounts of the different IFT proteins in the pool. The post-bleach gaps were similar for proteins within each subcomplex suggesting that IFT proteins at the flagellar base are organized into subcomplexes, rather than entire IFT particles. In detail, the data suggest that during the assembly of a given train IFT-A and B2 are recruited first, followed by addition of B1, and finally binding of the anterograde motor as visualized by KAP-GFP.

The sequential arrival of IFT subcomplexes in the bb-pool does not necessarily indicate that the recruitment of a late arriving subcomplexes depends on the presence of the earlier arriving ones. Knock-out of the IFT-A core protein IFT144, for example, prevents the assembly of functional IFT-A complexes but IFT-B proteins still enter cilia and accumulate at the tip due to the lack of retrograde traffic (*Hirano et al., 2017*). Thus, the sequential recruitment of IFT subcomplexes into the pool does not automatically reflect the interdependence during train assembly. We consider it more likely that IFT subcomplexes are recruited independently of each other and line up in (spatially) separated queues to be then combined into trains. To summarize, our data indicate a functional, spatial, and temporal organization of the IFT basal body pool.

## Tubulin is loaded briefly before the IFT trains enter the cilium

It has been suggested that axonemal proteins including tubulin and radial spoke subunits already associate to IFT proteins on IFT-coated post-Golgi vesicles and remain in this association during the arrival at the ciliary base, vesicle fusion with the plasma membrane, train formation, and transport into the cilium (*Wood and Rosenbaum, 2014*). To gain insights into the timing and location of cargo loading, we used sfGFP-tubulin, which due to its high frequency of transport by IFT is well suited for our photobleaching approach (*Craft et al., 2015*). The calponin-homology (CH) domain of IFT81 and the N-terminal domain of IFT74 bind tubulin in vitro, and cilia assembly is impaired in *C. reinhardtii* strains carrying mutations in both domains indicating that the IFT81/74 module is the dominant tubulin binding site of IFT (*Bhogaraju et al., 2013*; *Kubo et al., 2016*; *Taschner et al., 2016*; *Zhu et al., 2017b*). While these two proteins were not analyzed here, we consider it unlikely that pre-formed IFT81/74/tubulin complexes are incorporated into trains shortly before departure because IFT81 and IFT74 are core structural components of the IFT-B1 complex (*Lucker et al., 2005*; *Taschner et al., 2014*). If IFT-B1 proteins were already bound to tubulin at the time of their recruitment and incorporation into the trains, the gap in tubulin traffic after bleaching of the bb-pool should correspond in duration to that of the IFT-B1 proteins. However, the gap in sfGFP-tubulin traffic was actually shorter than that of KAP or any IFT particle protein studied here. The data indicate that tubulin loading occurs in the basal bodies pool and briefly before the trains move into the cilium. Axonemal cargoes are mostly released from IFT trains at the ciliary tip where IFT trains partially disassemble to reorganize for retrograde traffic (*Bower et al., 2013*; *Craft et al., 2015*; *Johnson and Rosenbaum, 1992*; *Lechtreck et al., 2013b*; *Wren et al., 2013*). We speculate that cargo binding and unloading could be linked to train assembly and disassembly.

## A model for the stepwise assembly of IFT trains

A critical role in train assembly can be attributed to the transitional fibers (TFs), which link the basal body triplets to the plasma membrane. *C. elegans* mutants in the TF protein Dyf-19 have severely reduced amounts of IFT proteins and IFT traffic inside cilia (*Wei et al., 2013*). The IFT-B1 protein IFT52 is associated with the TFs and the TFs are required to recruit kinesin-2, but not the IFT particle proteins, into the bb-pool (*Cole et al., 1998*; *Deane et al., 2001*). The entry of IFT proteins into the cilium depends on kinesin-2 (*Kozminski et al., 1995*), the binding of kinesin-2 to IFT is regulated by phosphorylation (*Liang et al., 2014*), and we show here that the addition of kinesin-2 is a late step of train assembly.

The longest average gap in IFT traffic was ~9s for the IFT-A/B2 proteins. *In C. reinhardtii*, IFT trains exit the bb-pool with a frequency of ~1/s suggesting that one train is completed each second and that ~9 trains in different stages of assembly are in the pool at a given time. This value fits with our estimates of the bb-pool size based on the fluorescence loss or gain upon departure/arrival of a single train. In a speculative model, each of the nine basal body triplets with its associated structures could assist to assemble one IFT train. The nascent trains move upwards along the triplet blades combining with additional IFT subcomplexes. Then, they will associate with kinesin-2 at the TFs, which will pull the nascent train upwards and into the cilium, compressing and concentrating the IFT material between the microtubules and the membrane. Once a position is vacated by a departing train, the assembly of a new train is initiated near the proximal end of the basal body (*Figure 8F*).

## Materials and methods

### Strains and culture conditions

The IFT20-FP (sfGFP or NG), IFT27-GFP, IFT43-YFP, KAP-GFP (CC-4296), and D1bLIC-GFP (CC-4488) strains were previously described; the corresponding strain numbers of the Chlamydomonas Resource Center (RRID:SCR_014960) are added in brackets when available. (*Lechtreck et al., 2009*; *Lv et al., 2017*; *Mueller et al., 2005*; *Qin et al., 2007*; *Reck et al., 2016*; *Zhu et al., 2017a2017*). KAP-GFP was expressed in the hypomorphic mutant *fla3* and IFT27-GFP was expressed in wild-type cells. The bald *ift46-1* mutant (CC-4375) has been previously described and was rescued by expressing IFT46-YFP (*Hou et al., 2007*; *Lv et al., 2017*). The novel *ift54* and *ift140 C. reinhardtii* mutants lacked flagella and were obtained by insertional mutagenesis using *aph7*, and *NIA1*, respectively, as selectable markers. PCR was used to confirm the insertions and specific antibodies confirmed the

loss of the wild-type proteins; details of the mutant strains will be reported in upcoming publications from the Lechtreck and Witman laboratories, respectively. A DNA fragment encompassing the coding region of IFT54 was amplified from genomic DNA and cloned downstream of NG into pBR25, which allows for selection on zeocin; the construct rescued the flagella-less phenotype of *ift54-2*. The *ift140* mutant was rescued using a construct consisting of the aphVIII selectable marker gene and the genomic region of IFT140 fused at its C-terminus to either sfGFP or mCherry codon-adapted for *C. reinhardtii*. The IFT140-sfGFP and –mCherry strains were grown in TAP medium, all other strains were maintained in modified M medium with a light/dark cycle of 14:10 hr. The *fla3* KAP-GFP *ift140* IFT140-mCherry strain was obtained by mating the *fla3* KAP-GFP and the *ift140* IFT140-mCherry strains. Motile progeny was analyzed by TIRF microcopy and the strains expressing KAP-GFP and IFT140-mC were analyzed by Western blotting using antibodies to C. reinhardtii KAP (Invitrogen, Carlsbad, CA) and IFT140 (*Picariello et al., 2017*). *Tetrahymena thermophila* expressing GFP-Dyf1p/IFT70 have been previously described (*Dave et al., 2009*). Briefly, the *GFP-Dyf1* cassette was inserted into the nonessential *BTU1* locus of a correspond knock-out strain; the cells were grown in SPP medium. A mIMCD3 cell line (RRID:CVCL_0429) stably expressing IFT88::YFP was a gift from Dr. Jagesh Shah (Harvard Medical School); cells were cultured as described (*Besschetnova et al., 2009*).

## TIRF microscopy

For TIRF imaging, we used Eclipse Ti-U microscope (Nikon) equipped with 60× NA1.49 TIRF objective and through-the-objective TIRF illumination provided by a 40 mW 488 nm and a 75 mW 561 nm diode laser (Spectraphysics) as previously described (*Lechtreck, 2013a*). The excitation lasers were cleaned up with a Nikon GFP/mCherry TIRF filter and the emission was separated using an Image Splitting Device (Photometrics DualView2 with filter cube 11-EM). Images were mostly recorded at 10 fps using an iXON3 (Andor) and the NIS-Elements Advanced Research software (Nikon); μManager (https://micro-manager.org/) was used to record some FLIP experiments.

To obtain a focused laser beam, the 488 nm laser beam was split using a 488 nm zero-order half-wave plate and a broad band polarized beam splitter; one of the beams was used for TIRF illumination. The other beam was expanded using a 3x beam expander, focused using 200 mm plano-convex lens and a 35 mm plano-convex lens and recombined with the TIRF laser beam using polarized beam splitter (all parts from Thorlabs Inc.). A motorized mirror connected to a joystick (Newfocus) was used to move the bleaching laser and the size of the laser spot was altered manually by moving the 35 mm lens. For FLIP experiments, the shutter and shutter driver (Uniblitz) for the bleaching laser were controlled via an Arduino Uno device and a custom-written macro for μ-Manager (see Supplementary methods). The FLIP experiments were recorded using either μ-Manger, which allowed us to prevent the acquisition of frames while the shutter was open or Nikon Elements; in the latter, the camera continued to record while the bleaching laser shutter was open resulting in over-exposed frames. FIJI (National Institutes of Health) was used to generate kymograms and quantify signals. Excel was used for statistical analysis; the kinetics of fluorescence recovery was determined manually based on the Excel scatter plots (see *Figure 2C* as an example). Adobe Photoshop was used to adjust image contrast and brightness, and figures were prepared in Adobe Illustrator. SigmaPlot in Tukey setting was used to prepare the box plots.

Observation chambers for *C. reinhardtii* were constructed by applying a ring of vacuum grease or petroleum jelly to a 24 × 60 mm No. 1.5 coverslip; 10 μl of cell suspension were applied and allowed to settle for ~1 min. Then, the chamber was closed by inverting a 22 × 22 mm no. 1.5 cover glass with ~5–10 μl of 5 mM Hepes, pH 7.3, 5 mM EGTA onto the larger coverslip. Cells were imaged through the large cover glass at room temperature.

TIRF microscopy of *T. thermophila* was performed as previously described (*Jiang et al., 2015*) with following modifications: Cells were washed and resuspended in Tris HCL buffer, pH 7.5, mixed in a 10:1 ratio with 20 μM $NiCl_2$ and applied to a 22 × 22 mm No. 1.5 cover glass. Nickel inhibits axonemal dyneins facilitating imaging of IFT (*Jiang et al., 2015*). A thin coat of petroleum jelly matching the outline of the cover glass was applied to a glass slide and inverted onto the cover glass. For in vivo imaging of IMCD cells expressing IFT88-YFP, the cells were grown under a transwell cup (Corning, Corning NY), switched to phenol red-free $CO_2$ independent medium with 25 mM HEPES (Gibco), and placed directly on a 22 × 60 mm No. 1.5 cover glass (*Ott and Lippincott-Schwartz, 2012*).

## FRAP and FLIP analysis of the basal body pool, IFT traffic, and tubulin transport

For FRAP analysis of the IFT bb-pool, kymograms were generated from videos showing the cells before, during and after the application of a brief laser pulse to bleach the experimental basal body. Grayscale profiles were plotted along lines covering the basal body signals, the data were converted to an Excel sheet, and the fluorescence of the experimental basal body was calculated in % of the intensity of the control basal body in a frame-by-frame manner. The recovery time and intensity were determined manually as shown in *Figure 2C*. In IMCD cells and *T. thermophila*, basal bodies with the attached cilium were bleached completely by moving the focused laser beam along the length of the cilium. To determine the rate of signal loss in FLIP experiments, we used the slope of a trendline added in Excel to the frames representing the initial 10–25 s of the experiment. For the gap analysis, a fiduciary mark on the monitor was used to position the cell so that one basal body was targeted by the laser spot; the focus level was adjusted to the level of the flagella, the recording was started, and a laser pulse of <100 ms to ~600 ms was applied. Cells which moved prior to the bleaching step or which did not resume regular IFT traffic after the bleaching step were ignored.

To analyze tubulin transport, cells expressing sfGFP-α-tubulin were washed and resuspended in M media, deflagellated by pH shock, transferred to fresh M medium, and allowed to regrow cilia under constant light with agitation [*Craft et al., 2015*; *Lefebvre, 1995*]). To delay the onset on regeneration, cells were kept on ice until needed. In stored cells, regeneration was initiated by diluting the cells with room temperature M media and incubation as above. For photobleaching of sfGFP-α-tubulin already present in the axoneme, the intensity of the 488 nm laser was increased from <1% to ~10% for 4–12 s.

To analyze the signal intensity of IFT trains, a line partially covering the trajectory was placed on kymograms in ImageJ and the grey value was obtained.

## Program used to control the laser shutter in μManager via an Arduino device

```
/* LaserGateLoop3.bsh
* This program will take a series of images after exposing a laser gate for a set
duration
* There is a pause or a user set amount of time in beween two loops.
* The first image is before the gate pulse
* The shutter trigger is Arduino pin 13 which should be connected to the Pulse Input
of
* the Uniblitz shutter. The shutter should be set to N.O. and Remote
* Heather Bomberger Summer 2014 University of Georgia
*/
gui.clearMessageWindow();
//USER INPUT
float ImageDuration = 800; //Time beween flashes (ms)
GatePulse = 100; //Time laser is on (ms)
int LoopOne = 35; //Number of flashes before the pause int LoopTwo = 35; //Number of
flashes after the pause
Pause = 10; //Duration of pause (ms) (The ImageDuration will be added to this
pause)
EMgain = 151; //Gain
Exposure = 100; //Exposure (ms)
//User Inerface (Delete "/*" and "*/" to use)
/* import ij.gui.GenericDialog;
GenericDialog gd = new GenericDialog("LaserGateLoop");
gd.addNumericField("Exposure", Exposure, 0);
gd.addNumericField("EM Gain", EMgain, 0);
gd.addNumericField("Gate Pulse", GatePulse, 0);
gd.addNumericField("Image Duration", ImageDuration, 0);
gd.addNumericField("Loop One", LoopOne, 0);
```

```
gd.addNumericField("Loop Two", LoopTwo, 0);
gd.addNumericField("Pause", Pause, 0);
gd.showDialog();
Exp = (int) gd.getNextNumber();
Gain = (int) gd.getNextNumber();
GatePulse = (int) gd.getNextNumber();
ImageDuration = (int) gd.getNextNumber();
LoopOne = (int) gd.getNextNumber();
LoopTwo = (int) gd.getNextNumber();
Pause = (int) gd.getNextNumber();
print (Exp); print (Gain);
print (GatePulse); print (ImageDuration);
print (LoopOne); print (LoopTwo);
print (Pause);
*/
//Check Statements
if (Exposure >1000|| Exposure <10) {
    gui.message("Exposure not in range.");
    break;
}
if (EMgain >300|| EMgain <0) {
    gui.message("EMgain not in range.");
    break;
} if (ImageDuration >5000 || ImageDuration <0) {
    gui.message("Image Duration not in range.");
    break;
} if (GatePulse > 500 || GatePulse < 0) {
    gui.message("Gate Pulse not in range.");
    break;
} if (LoopOne >100 || LoopOne <0) {
    gui.message("Loop One not in range.");
    break;
} if (LoopTwo > 100 || LoopTwo < 0) {
    gui.message("Loop Two not in range.");
    break;
} if (Pause >5000 || Pause <0) {
    gui.message("Pause not in range.");
    break;
}
Camera = mmc.getCameraDevice();
mmc.setProperty(Camera, "Exposure", Exposure);
mmc.setProperty(Camera, "Gain", EMgain);
Interval = mmc.getProperty(Camera, "ActualInterval-ms");
FloatInterval = Float.valueOf(Interval);
int PauseFrames = (int) (Pause/FloatInterval);
print("Interval: " +FloatInterval); print(ImageDuration / FloatInterval);
mmc.setProperty("Arduino-Switch", "State", 32);
mmc.setProperty("Arduino-Shutter", "OnOff", 0);
mmc.setAutoShutter(false);
//auto shutter off gui.enableLiveMode(false);
//live mode off int nrFrames = (int) (ImageDuration / FloatInterval);
int frame = 0;
int nrImages = nrFrames*(LoopOne+LoopTwo)+1 + PauseFrames;
int imgcnt = 0; acqName = "Test";
rootDirName = "C:/Users/Lechtreck Lab/Documents/Heather-MM";
nrChannels = 1;
```

```
nrSlices = 1;
int channel = 0;
int slice = 0;
width = mmc.getImageWidth();
height = mmc.getImageHeight();
bytes = mmc.getBytesPerPixel();
depth = mmc.getImageBitDepth();
print ("width: " +width);
print ("height: " +height);
print ("depth: " +depth);
print ("bytes: " +bytes);
print ("Frames: " +nrFrames);
gui.closeAllAcquisitions();
gui.openAcquisition(acqName, rootDirName, nrImages, nrChannels, nrSlices);
gui.initializeAcquisition(acqName, (int) width, (int) height, (int) bytes,
(int) depth);
//gui.setAcquisitionProperty(acqName, "Pixel width", "2.0000");
//mmc.setProperty("Properties", "Pixel width", 2);
now = System.currentTimeMillis();
// Acquire a pre-bleaching image and add it to the acquisition at the beginning
mmc.snapImage();
PreImage = mmc.getImage();
gui.addImage(acqName, PreImage, frame, channel, slice);
imgcnt++; //the acquisition will start on frame 2
gui.setContrastBasedOnFrame(acqName, frame, slice −1);
//laser gate for loop One
for (n = 0; n<LoopOne; n++){
    frame = 0;
    mmc.setProperty("Arduino-Shutter", "OnOff", 1);
    mmc.sleep(GatePulse);
    mmc.setProperty("Arduino-Shutter", "OnOff", 0);
    mmc.startSequenceAcquisition(nrFrames, 0, false);
    while (mmc.getRemainingImageCount()>0 || mmc.isSequenceRunning(mmc.getCa-
meraDevice()))
{
      if (mmc.getRemainingImageCount()>0) {
        img = mmc.popNextImage();
      if (frame<nrFrames) {//take designated amount of images
      gui.addImage(acqName, img, imgcnt, channel, slice);
        frame++;
        imgcnt++;
    }
      }
  }
}
//Pause while camera recording
    frame = 0;
    mmc.startSequenceAcquisition(PauseFrames, 0, false);
    while (mmc.getRemainingImageCount()>0 || mmc.isSequenceRunning(mmc.getCa-
meraDevice()))
{
      if (mmc.getRemainingImageCount()>0) {
        img = mmc.popNextImage();
      if (frame<PauseFrames) {//take designated amount of images
      gui.addImage(acqName, img, imgcnt, channel, slice);
        frame++;
```

```
        imgcnt++;
      }
    }
  }
//Loop Two
for (n = 0; n<LoopTwo; n++){
   frame = 0;
   mmc.setProperty("Arduino-Shutter", "OnOff", 1);
   mmc.sleep(GatePulse);
   mmc.setProperty("Arduino-Shutter", "OnOff", 0);
   mmc.startSequenceAcquisition(nrFrames, 0, false);
   while (mmc.getRemainingImageCount()>0 || mmc.isSequenceRunning(mmc.getCa-
meraDevice()))
{
   if (mmc.getRemainingImageCount()>0) {
     img = mmc.popNextImage();
   if (frame<nrFrames) {//take designated amount of images
   gui.addImage(acqName, img, imgcnt, channel, slice);
     frame++;
     imgcnt++;
   }
    }
  }
}
duration = (System.currentTimeMillis() - now);
gui.setContrastBasedOnFrame(acqName, frame, slice -1);
mmc.stopSequenceAcquisition();
gui.promptToSaveAcquisition(acqName,false);
gui.closeAcquisition(acqName);
print ("Duration = " +duration +" ms");
mmc.setProperty("Arduino-Shutter", "OnOff", 0);
```

## Acknowledgements

We are grateful to Mary Porter (University of Minnesota), Wallace Marshall (UCSF), Kaiyao Huang (Institute of Hydrobiology, Chinese Academy of Sciences), and Jagesh Shah (Harvard) for providing strains expressing FP-tagged IFT proteins. We thank Ahmet Yildiz (Berkeley) for advice on how to install a focused bleaching laser beam and Philippe Bastin (Institut Pasteur) for fruitful discussion. Heather Bomberger was supported by the NSF-funded REU program 'Nanotechnology and Biomedicine' (NSF EEC-1659525 and −1359095). Research reported in this publication was supported by the Robert W. Booth Fund at the University of Massachusetts Medical School (GW) and by the National Institute of General Medical Sciences of the National Institutes of Health under award numbers R37GM030626 (GW) and R01GM110413 (KL). The content is solely the responsibility of the authors and does not necessarily represent the official views of the National Institutes of Health. The authors declare that they have no conflict of interest.

## Additional information

### Funding

| Funder | Grant reference number | Author |
|---|---|---|
| National Institutes of Health | General Medicine - R01GM110413 | Karl Lechtreck |
| Korea National Institute of Health | General Medicine - R37GM030626 | George B Witman |

The funders had no role in study design, data collection and interpretation, or the decision to submit the work for publication.

## Author contributions

JLW, Y-YJ, JDW, Investigation; IM, JMB, TP, DAC, BZ, JP, Resources; HB, PK, Software, Methodology; JE, Resources, Supervision; JG, Resources, Supervision, Writing—review and editing; GBW, Resources, Writing—review and editing; KL, Conceptualization, Investigation, Writing—review and editing

## Author ORCIDs

Karl Lechtreck, http://orcid.org/0000-0002-6219-6470

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
