## [Decision Letter]

Thank you for submitting your article "IFT trains in different stages of assembly queue at the flagellar base for consecutive release into cilia" for consideration by *eLife*. Your article has been favorably evaluated by Anna Akhmanova (Senior Editor) and three reviewers, one of whom, Erwin J G Peterman (Reviewer #1), is a member of our Board of Reviewing Editors.

The reviewers have discussed the reviews with one another and the Reviewing Editor has drafted this decision to help you prepare a revised submission.

Summary:

The process of intraflagellar transport is a complex intracellular transport system critical for assembly of cilia and flagella, organelles involved in a wide range of physiological and developmental processes. The mechanisms by which cells assemble such a complex structure as the cilium remain poorly understood, and IFT provides an excellent system in which to study the interplay between assembly and transport. The IFT system is itself highly complex, raising the question of how IFT particles and trains assemble and disassemble as they journey between the cell body and cilium. In this manuscript, which is a tour de force of live-cell imaging, the authors address the assembly of IFT particles, and the dynamics of their exchange between intracellular compartments, at an unprecedented level of molecular and temporal resolution.

By combining photobleaching with a battery of fluorescent protein tagged IFT constructs, the authors are able to reach several important conclusions. First, they find that unlike the situation reported for Trypanosomes, IFT in *Chlamydomonas* seems to be an open system in which retrograde particles are returned to a general cytoplasmic pool rather than being sequestered in a flagellum-basal body system. Second, they are able to show that the size of a basal body-associated IFT pool is small relative to the total cellular pool. Further quantitative analysis of photobleaching and FLIP experiments allowed the authors to calculate how much of the basal pool is regenerated from retrograde particles, and how much of the pool is released for each anterograde train. Finally, using different constructs they are able to estimate the time-line of IFT particle assembly, for example showing that motors assemble relatively late in the process, and cargo even later. This hints at a stepwise assembly process and provides critical in vivo kinetic information to complement some of the biochemical associate studies currently underway in other labs.

All three reviewers agree that the design of the experiments is clever, that the quality of the data is very good, that quantification is (in most cases) reliable, that appropriate and clever control experiments have been performed (e.g. evidence that the fluorescent proteins are functional, use of the basal body of the second flagellum as FRAP/FLIP controls, recovery of normal IFT after FLIP removal) and that the manuscript is clear and reads well. After the authors have addressed the following comments, we all three support publication of this important work in *eLife*, since it provides important novel insights that will not only be exciting for the cilia community, but also the wider cell biology and protein trafficking communities.

Essential revisions:

1) Subsection “IFT proteins occupy distinct regions of the basal body pool”: Relative localisation of IFT proteins and motors in the basal pool. The differences shown are close to the resolution limits of the microscopy method used. It is wise to be a bit more careful to draw conclusions, unless double staining data (like in Figure 7) or super-resolution data (SIM, STED) is available.

2) Figure 2. It is not clear how recovery times were determined. Was this done with fitting? Please explain and show data (and fits!). Figure 2. Given the relatively small N's (which is fine in itself), the histograms could be represented as box plots, including all data points, which might give insights in the underlying distributions.

3) Figure 3. This figure is a bit weird: the error bars of some data entries are larger than the average values. What is going on here? Are there negative data points? Please also show all data points here in a boxplot.

4) Subsection “The dispatched IFT-A and motor proteins do not return to the basal body pool”. The statement (15s) appears inconsistent with Figure 3—figure supplement 1, where plateau was reached after ~50s. How was this determined? With a fit? Please explain and show data.

5) Subsection “Anterograde trains are largely assembled from proteins freshly recruited to the basal bodies”, last paragraph: "After 6 minutes of FLIP the IFT pool was largely exhausted". Can it be excluded that photobleaching due to imaging (and *not* FLIP) plays a role here?

6) Figure 6. The average gap time before visible anterograde particles is used to infer differences in loading time of different IFT proteins. I would suggest that the authors also show us the actual distribution in gap times. This might give insight in the mechanism of assembly (in case every protein is incorporated independently, one would expect exponential distributions, in case there are distinct intermediates, one might expect γ-distributions). The gap times are now interpreted as representing the assembly times. However, it could also reflect the abundance of a given fluorescent protein at the basal body pool. This could be lower for motors and IFT-A than for IFT-B and could explain the results (faster recovery of the signal at the base, faster injection of fluorescent trains). This would be difficult to quantify precisely since different expression systems and different fluorescent proteins were used; this deserves, however, discussion. In addition, the stoichiometry of tubulin and IFT trains is unknown: tubulin-binding domains have been found on several IFT proteins (Bhogaraju et al. 2014). Please discuss how this complexity could affect the data.

7) The discovery of open and semi-open IFT system adds up to the already known apparently closed systems of *T. brucei* and *C. elegans*. These are important results and their biological significance deserves to be discussed in a specific paragraph.

---

## [Author Response]

*Essential revisions:*

*1) Subsection “IFT proteins occupy distinct regions of the basal body pool”: Relative localisation of IFT proteins and motors in the basal pool. The differences shown are close to the resolution limits of the microscopy method used. It is wise to be a bit more careful to draw conclusions, unless double staining data (like in Figure 7) or super-resolution data (SIM, STED) is available.*

We agree that super-resolution approaches are likely to provide a much more detail map of the basal body pool. To acknowledge the limitations of our imaging technique, we added:

“Within the limitations of the technique used, live cell imaging confirms and extends pervious observations on fixed cells indicating that different IFT proteins inhabit non-identical spatial domains within the basal body pool (Hou et al., 2007).”

And in the Discussion:

“Super-resolution imaging is likely to provide a clearer image of IFT protein territories within the basal body pool” (referred to as bb-pool in the revised manuscript).

It should be noted that many of our observations are backed-up by published immunofluorescence data and that the XY resolution of TIRF is surprisingly good, as shown in Figure 5.

*2) Figure 2. It is not clear how recovery times were determined. Was this done with fitting? Please explain and show data (and fits!). Figure 2. Given the relatively small N's (which is fine in itself), the histograms could be represented as box plots, including all data points, which might give insights in the underlying distributions.*

Recovery times etc. were determined manually based on the Excel scatter plots. This has been added to the Materials and methods section and a representative example is shown in Figure 2 and explained in the figure legend.

Materials and methods section:

“Excel was used for statistical analysis; the kinetics of fluorescence recovery was determined manually based on the Excel scatter plots (see Figure 2 as an example).”

Legend of Figure 2:

“The time needed for recovery was determined manually by determining the point of interception of lines along the slope during recovery and the plateau of the recovered signal (red dashed lines).”

The bar diagram in Figure 2 was replaced with a box plots including the data points.

*3) Figure 3. This figure is a bit weird: the error bars of some data entries are larger than the average values. What is going on here? Are there negative data points? Please also show all data points here in a boxplot.*

This panel summarizes our FLIP experiments from which we conclude that IFT-A and motor proteins do not return to the pool after passing through the cilium. In agreement with this notion, the signal of the experimental basal body at the end of the experiment can be somewhat stronger or weaker than that of the control basal body. Thus, negative data points are to be expected.

Due to the large spread of data points (~ -20 – 60), the differences between the three IFT-B proteins and the other proteins are difficult to appreciate in the box plot presentation. We therefore kept the bar diagram in the main figure and added the corresponding box plot of the data as a supplementary figure (Figure 3—figure supplement 1, panel E).

*4) Subsection “The dispatched IFT-A and motor proteins do not return to the basal body pool”. The statement (15s) appears inconsistent with Figure 3—figure supplement 1, where plateau was reached after ~50s. How was this determined? With a fit? Please explain and show data.*

In the example in Figure 3—figure supplement 1, the experimental basal body pool indeed loses its signal quite slowly. While we do not know why this particular cell behaves this way, possible reasons for such a slow rate of signal loss are insufficient strength of the bleaching laser or incorrect position of the laser with respect to the ciliary tip as it can result from small cell or flagella movements. Both would allow for some retrograde trains to return to the bb-pool delaying the loss of fluorescence. Such residual retrograde IFT would not have been out of focus in such recordings such as the one shown here. We replaced this panel with a kymogram from a more typical recording.

To quantify the loss of signal in the basal body pool in FLIP experiments, we recorded the cells using micromanager (and our macro that to avoid recording while the bleaching laser was on; see supplement). A quantification of such a recording is shown in Figure 3 (original and revised manuscript). For this set of experiments we mostly ensured that retrograde traffic was abolished by changing the focus level toward the end of the recording to check for residual retrograde traffic (as shown in Figure 3). The rate of signal loss in FLIP experiments was determined as follows (the sentence was added to the Materials and methods section):

“To determine the rate of signal loss in FLIP experiments, we used the slope of a trendline added in Excel to the frames representing the initial 10 – 25 s of the experiment.”

Time to reach the plateau and level of plateau: For NG-IFT54, we provide box plots of the time required to reach the plateau and the signal strength at the plateau (Figure 2—figure supplement 1). The data were analyzed by manual inspection of the scatter plots.

*5) Subsection “Anterograde trains are largely assembled from proteins freshly recruited to the basal bodies”, last paragraph: "After 6 minutes of FLIP the IFT pool was largely exhausted". Can it be excluded that photobleaching due to imaging (and not FLIP) plays a role here?*

Yes, at least for mNeonGreen-tagged IFT54 we can exclude that the loss of signal is simply due to bleaching caused by the observation light. mNG is very bright and photostable allowing us to image the cells at low laser intensities (avoiding photobleaching) for extended periods of time.

In the original (and revised) manuscript we stated:

“In control cells imaged for similar periods of time without being directly hit by the FLIP laser, dense IFT traffic albeit of reduced intensity persisted (not shown).”

Since this question is listed as an essential revision, we added the kymogram of a control cell (from a recording using the same settings of the lasers for bleaching and recording as the experimental cells) as a new panel D in Figure 3—figure supplement 2 to the revised manuscript.

(“In control cells imaged for similar periods of time without being directly hit by the FLIP laser, dense IFT traffic albeit of reduced intensity persisted (Figure 3—figure supplement 2).”)

*6) Figure 6. The average gap time before visible anterograde particles is used to infer differences in loading time of different IFT proteins. I would suggest that the authors also show us the actual distribution in gap times. This might give insight in the mechanism of assembly (in case every protein is incorporated independently, one would expect exponential distributions, in case there are distinct intermediates, one might expect γ-distributions).*

Box plots with data points have been included in the revised Figure 6.

*The gap times are now interpreted as representing the assembly times. However, it could also reflect the abundance of a given fluorescent protein at the basal body pool. This could be lower for motors and IFT-A than for IFT-B and could explain the results (faster recovery of the signal at the base, faster injection of fluorescent trains). This would be difficult to quantify precisely since different expression systems and different fluorescent proteins were used; this deserves, however, discussion.*

We agree that this is an important point, which was insufficiently considered in the original manuscript. We added the following to the Discussion:

“The data imply that the bb-pool should contain more IFT-A proteins (in equivalents of IFT trains) than KAP. The use of different FPs, the distinct spatial distribution of IFT protein in the bb-pool, and the expected small scale of the expected differences prevented us from comparing the amounts of the different IFT proteins in the pool.”

Our conclusion that proteins (of a given train) are sequentially recruited to the pool implies that the respective pools will be different in size as well (larger for IFT-A and smaller for the motors); for the IFT-B proteins, which also return to the pool, the situation is more complicated.

*In addition, the stoichiometry of tubulin and IFT trains is unknown: tubulin-binding domains have been found on several IFT proteins (Bhogaraju et al. 2014). Please discuss how this complexity could affect the data.*

The theoretical work of Bhogaraju et al. 2014 has been superseded by Taschner et al. (2016) showing that only the CH domains of IFT81 and IFT54 bind tubulin and microtubules. Further, while our manuscript was under revision, Zhu et al. (2017) showed that the CH domain of IFT54 is not required for flagellar assembly in *Chlamydomonas*. In addition, Kubo et al. (2016) showed that a combination of mutations in the CH of IFT81 with an N-terminal truncation of IFT74 (IFT81’s binding partner) renders *Chlamydomonas* cells unable to assemble flagella. Thus, tubulin binding sites independent of the IFT81/74 module appear to be absent in *Chlamydomonas*. To address this point we updated the references and rewrote the corresponding part of the Discussion as follows:

“To gain insights into the timing and location of cargo loading, we used sfGFP-tubulin, which due to its high frequency of transport by IFT is well suited for our photobleaching approach (Craft et al., 2015). […] While these two proteins were not analyzed here, we consider it unlikely that pre-formed IFT81/74/tubulin complexes are incorporated into trains shortly before departure because IFT81 and IFT74 are core structural components of the IFT-B1 complex (Lucker et al., 2005; Taschner et al., 2014).”

*7) The discovery of open and semi-open IFT system adds up to the already known apparently closed systems of T. brucei and C. elegans. These are important results and their biological significance deserves to be discussed in a specific paragraph.*

We extended our discussion on the differences in the IFT systems of *Chlamydomonas, T. brucei*, and *C. elegans* by adding the following two phrases to the corresponding section of the Discussion:

“For example, in a semi-open system as described for the IFT-B proteins, the ratio between new recruitment from the cb-pool and reusage of proteins derived from retrograde trains could be different between species with *T. brucei* favoring reusage.”

And

“In their mature state, *C. elegans* cilia possess altered basal bodies and lack the transitional fibers, two structures that are likely to be critical for the recruitment of IFT proteins from the cb-pool to the ciliary base; these changes in could favor a more closed IFT system (Nechipurenko et al., 2017)*.”*